

**Impact of Cropping Systems on Macronutrient Distribution and Microbial Biomass in**
**Drought Affected Soils.**
M. Naga Jayasudha[1], M. Kiranmai Reddy[1,2], Surendra Singh Bargali[3]
1. Department of Environmental Science, School of Sciences, 2. Multi-disciplinary Unit of
Research on Translational Initiatives (MURTI), GITAM (Deemed to be University),
Visakhapatnam-530045, Andhra Pradesh.
3. Department of Botany, D.S.B. Camus, Kumaun University, Nainital -263001, Uttarakhand,
India
Corresponding author: kmajji@gitam.edu
**Abstract**
The interplay between soil nutrients, water activity, and microbial biomass is pivotal for plant growth
as well as for soil health. While surface microflora typically promotes mineralization and nutrient
deposits, the impact of drought on soil microbial biomass and nutrient utilization remains
underexplored. In this study, we assessed various land types—open lands (OL), annual crops with single
species (ACS), perennial crops with multiple species (PCM), less water available lands (LWA), and
soil near ponds (CP)—to elucidate the distribution of macronutrients and microbial biomass. Soil
samples were collected from different land types, air-dried, and subjected to physical, chemical, and
biological analyses. Standardized protocols, including gravimetric and titration analyses, were
employed for physical and chemical assessments, while microbial biomass was evaluated using
fumigation. Statistical analyses, including ANOVA and Pearson Coefficient, were employed to discern
patterns across seasons, soil depths, and microbial biomass. Microbial biomass carbon (Cmic) ranged
from $134.2\pm1.2\mu g/g$ to $286.6\pm1.33\mu g/g$, while nitrogen (Nmic) and phosphorus (Pmic) varied from
$11.3\pm1.3\mu g/g$ to $69.5\pm0.98\mu g/g$ and $07.6\pm1.5\mu g/g$ to $77.5\pm0.6\mu g/g$, respectively, across all seasons.
Carbon stock in the upper soil surface positively influenced nitrogen and phosphorus retention. Notably,
PCM exhibited superior Cmic, Nmic, Pmic, and water-holding capacity compared to OL, LWA, and
ACS. Our findings underscore the significance of multiple cropping systems, particularly PCM, in
enhancing microbial biomass and nutrient levels in drought-affected regions. The observed
improvements in soil moisture, nitrogen, phosphorous, and potassium levels suggest that diverse
cropping systems can effectively enrich soil nutrients and biomass content in drought stress. In
conclusion, our study highlights the potential of perennial crops with multiple species in mitigating the
impact of drought on soil microbial biomass and macronutrient distribution. These findings contribute
to a deeper understanding of sustainable agricultural practices in drought-prone regions and emphasize
the importance of implementing diverse cropping systems to enhance soil health and resilience.
**Key words:** Microbial biomass. Soil properties. Cropping systems. Drought region.
**Introduction**



One of the important indicators of the soil is microbial biomass, which plays a crucial role in
maintaining organic content in the soil by decomposing organic matter and hence controlling
the nutrients and maintaining the biogeochemical process of different ecosystems (Wang et al.,
2014; Manral et al., 2020). The microbial density and stability are greatly affected by the
different ecosystems and the existence of nutrient supplements in the soil (Dietterich et al.,
2022; Manral et al., 2023). The present agricultural practices are in high demand for using
pesticides, fertilizers, and hybrid seeds at a very high rate, which results in environmental
degradation, particularly soil. The productivity of crops mostly depends on the existence of
nutrients in the soil that later reflect on the contribution made by the soil microflora in terms
of microbial biomass. Amendments of organic contents to the soil provide nutrients that help
in the colonization of microbial communities (Bastida et al., 2017), thereby, changes in soil
characteristics might be noticed.
Bulk density is directly related to soil compaction. In the open land use system, the bulk density
is higher because of the soil compaction (Bargali et al., 1993; Joshi et al., 1997). This is also
related to soil microbial activities; due to more soil pore space; moisture supports the microbes
to enter the available pore space and enhance their activities and soil becomes more porous in
forested land (Bargali K et al., 2018). Vegetation plays a significant role in the formation of
soil organic matter (SOM) and influences fundamental soil-forming processes such as
aggregation or podzolization (Awasthi et al., 2022 a & b). Han et al., (2021) stated that floristic
composition plays an important role in the formation of SOM and influences fundamental soil-
forming processes. The texture of the soil may also affect the productivity of the forest by
affecting moisture availability and nutrient supply to microbial decomposition (Bargali et al.,
2015; Han et al., 2021).
The occurrence of drought might be due to changes in the land use patterns, which change
nature by physical, chemical, and biological means, which modifies the soil properties and
increases erosion and level of compaction (Geissen et al., 2009; Maranguit et al., 2017). The
supply of nutrients in the soil is due to the weathering of rocks and is further processed by the
decomposition of the organic matter, which results in different forms as organic and inorganic,
available and non-available; however, the carbon, nitrogen, and phosphorus concentration in
the soil enriches through microbial activity (Chen et al., 2022). The management of cropping
patterns maintains the interaction between soil and microbes by the addition of organic matter
either by physical change or by nutrient supply to the soil (Devi & Yadava, 2006). It has been
shown that the plantation of mixed tree species has influenced the composition of the aerobic



and nitrifying bacterial communities in the soil and has helped mitigate the drought effects in
the cropping system (Gillespie et al., 2020). Earlier, it has been demonstrated that soil cropping
practices enhance soil's carbon, nitrogen, and phosphorus content and could increase the
diversity of soil microbes, enhancing the tolerance to abiotic stresses (Bastida et al., 2017).
Therefore, restoring the soil microbial communities, maintaining nutrient cycling, and
promoting effective crop productivity is of utmost necessity in drought regions.
Assessing the soil microbial biomass to maintain soil health provides long-term productivity
in different cropping patterns. Studies in this line about the different cropping systems and their
microbial biomass in drought regions have not been carried out earlier. Furthermore, its
relationship with nutrients was also not studied. Hence, an attempt has been made to understand
the microbial biomass of carbon, nitrogen, and phosphorus with different cropping systems in
the drought region of Andhra Pradesh. Here, a study has been conducted to understand the
influences of multiple cropping systems on soil microbial biomass in drought-hit regions in
terms of soil depths, seasonal variation, nutrient composition, and diversity. Our findings
suggested that multiple cropping systems have helped enhance the soil microbial biomass and
macronutrients like carbon, nitrogen, and phosphorous in drought-hit soils.

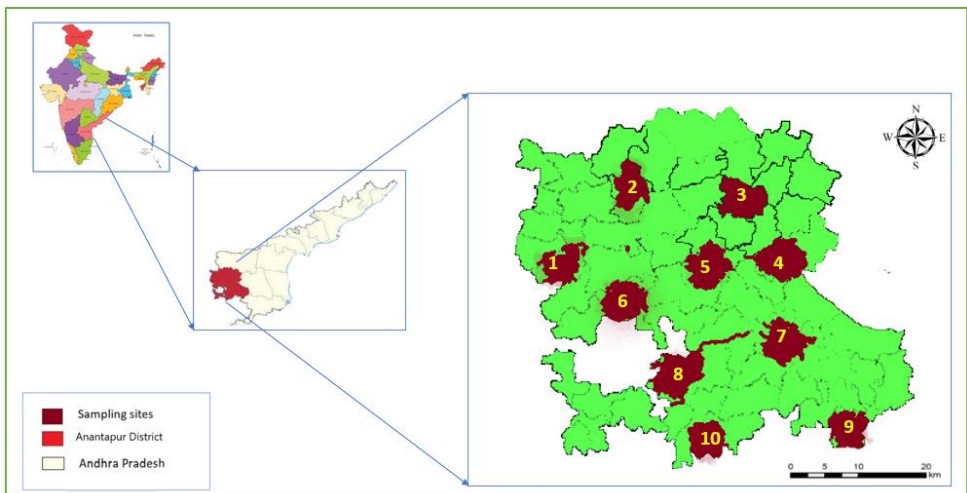

Fig-1 Selected ten sampling sites for the soil analysis of the Ananthapuram district of Andhra
Pradesh, India.
**2 Material and Methods**
**2.1 Study area and Climatic conditions**



The present study was in the drought-hit region of Anantapur district of Andhra Pradesh with a 14.67° and 14. 80° N, 77.63°, and 78.16°E with an average elevation of 335 m. In the study, the cropping pattern has been categorized as open lands (OL), Annual Crops with single species (ACS), Perennial crops with multiple species (PCM), Less water available lands (LWA), and Crops grown near the ponds (CP) has been selected as control. The pattern of the cropping system has been described in Table 1. The average climatic conditions of the area are semi-arid, with hot and dry conditions and occasional rainfall. The whole is comprised of three seasons, summer with an average temperature of 36° C (March to June), Monsoon (July to October) with a temperature of 31° C, and winter season lasts for three months (November to February) with an average temperature of 18° C.

**2.2 Collection of Soil Samples**

From the study site, ten samples from 10 different regions were collected randomly from three different seasons, viz., summer, monsoon, and winter, with different soil depths such as upper surface (0-15 cm), subsurface (15-30 cm) and deeper layers (30-45 cm), after collecting soil samples in Ziplock bags, they were transported to the laboratory and air dried. The samples were made into three sub-samples, and different soil characteristics were analyzed.

**2.3 Soil Analysis**

The physical parameters such as bulk density (BD), soil moisture (SM), water holding capacity (WHC), soil texture, and soil temperature (ST) were analysed (Misra 1968). The texture was analysed with different measurements of the sieve size. The soil's chemical properties are pH, electric conductivity (EC), total organic carbon (Walkley and Black 1934), soil organic matter, total nitrogen (Peach & Tracy 1956), phosphorus (Olsen et al., 1954), and potassium (Pratt 1965). Soil microbial analysis was estimated by taking the surface soils as the activity of microbes will be higher in the surface soils. The chloroform fumigation method is used to evaluate $C_{mic}$, $N_{mic}$, and $P_{mic}$ (Brookes et al., 1985; Vance et al., 1987).

**2.4 Statistical Analysis**

The statistical analysis was carried out by using ANOVA to understand the impact of seasons on the different agrosystems and their interactions with soil nutrients particularly, $C_{mic}$, $N_{mic}$, $P_{mic}$. Pearson's correlation matrix was established for the data collected using SPSS version 16.

**3 Results**



### 3.1 Physicochemical characteristics

The information about the physicochemical characteristics of the soil has been tabulated in Table-2 and Table-3. The agricultural systems have a similar texture, indicating that the soil has been derived from the same parental rock, which underwent several physical, chemical, and biological weathering; moreover, due to different management practices, its original characteristics could be changed. The highest sand percentage has been observed in LWA, silt in CP, and clay in PCM. When we look at bulk density, it was seen higher in OL with 2.01 g/cm$^3$ when compared to CP with 1.65 g/cm$^3$, indicating that it is more than 10% higher in open lands due to cropping systems. A significant variation has been observed in the WHC parameter, where PCM was identified at 51.2% with the highest and CP with the lowest at 33.24%. The soil temperature ranged from 29.3°C (LWA) to 32.1°C (OL), whereas the soil moisture recorded from 4.08% to 9.43% in all agricultural cropping systems.

The pH is an important chemical characteristic of soil that decides the soil acidity or alkalinity for crop production. It is identified in ACS as 8.93, and others with a range of 7.79 to 8.66. The increase in the pH clearly states the enhanced use of synthetic fertilizers and pesticides in the soil. The dissipation of ions in the soil is also confirmed by the EC study of the different ranges of the cropping pattern, which are in the series of 0.13dS/m in ACS to 0.38dS/m in OL. The variation influences the distribution of ions in the soil. The other chemical parameter in the study which promotes good crop productivity is organic carbon, which rangesfrom 0.24% (LWA) to 0.91% (OL), mostly found in the surface soils compared to subsurface soils.

The soil nitrogen has also shown similar with surface soils having higher concentrations with a range of 1216 kg/ha (LWA) to 1354 kg/ha (OL); in the case of phosphorus 10 kg/ha was observed in LWA whereas 21 kg/ha has been identified in PCM. The potassium levels ranged from 132 kg/ha (LWA) to 432 kg/ha (PCM). The variation in the NPK availability was higher than less water available lands, which are mostly considered discarded lands. Most of these cultivated lands have shown nitrogen with 90% and above compared to low water level lands due to the continuous addition of synthetic fertilizers. The difference might be due to the external addition of fertilizers reducing the loss of nitrogen in the cultivated lands.

### 3.2 Carbon ($C_{mic}$), Nitrogen ($N_{mic}$), Phosphorus ($P_{mic}$) microbial biomass

In the study, the microbial biomass of carbon-nitrogen and phosphorus has shown higher values for perennial crops with multiple species when compared to other crop soils (Table-4). Throughout the cropping systems, the microbial biomass carbon registered with a range of



177.6±0.89 in OL to 286.6±1.33 in PCM. In the case of $N_{mic}$, the concentration has ranged from
43.5±0.77 (LWA) to 69.5±0.98 (PCM), whereas the $P_{mic}$ has shown a variation of about
08.7±0.67 in the dry season of CP to a maximum of 77.5±0.6 in the monsoon season of PCM.
This clearly indicates the accumulation of organic carbon in perennial crops compared to other
cropping systems. The microbial biomass has shown change (Table-5) due to different
cropping patterns and seasonal variations. The microbial biomass variation in the soil might be
due to the patterns of crops and their interactions with soil, and the exchange of the materials
could cause fluctuations in the biomass content. When compared to the $C_{mic}$, both $N_{mic}$ and
$P_{mic}$ have shown more concentration, probably due to the interaction of microorganisms to
nitrogen and phosphorus being quite quick than carbon, as these are basic requirements for the
microbes for the growth stages. In these cropping systems, perennial cropping soils have better
physicochemical and biological parameters due to the constituent of the soil, well root
structure, and plant litter throughout the year, making the availability of microbial biomass
carbon, nitrogen, and phosphorus. Whereas the lowest $C_{mic}$, $N_{mic}$, and $P_{mic}$ were found in OL
followed by CP due to fewer crops left in these soils, the concentration of C:N and C:P ranged
from 11.5±1.9 to 17.2±1.8 and 12.6±1.1 to 22.1±1.4 as marked in Table-4.
**3.3 Physicochemical characteristics and their relationship with microbial biomass**
Pearson's correlation between microbial biomass and physicochemical characteristics is given
in Table-6. The microbial biomass carbon showed a significant positive correlation with $N_{mic}$
(r=0.69) and $P_{mic}$ (0.234), pH (r=0.55), Organic Carbon (r=0.743), WHC (r=0.789), SM
(r=0.665), N(r=0.489), P (r=0.599), K (r=0.564) and have a negative correlation with sand
contents. In the case of $N_{mic}$, a positive correlation has been noticed with $P_{mic}$ (r=0.576), pH
(0.63), Organic carbon (0.853), WHC (r=0.493), BD(2.55), SM (0.665), N(r=0.756),
P(r=0.486), K(r=0.564), whereas $P_{mic}$ also showed a significant positive correlation with these
physicochemical parameters and negative with sand contents as shown in table-6.
**4 Discussion**
Soil ecology, nutrients richness, and other physical, chemical, and biological properties
determine the agriculture sustainability and cropping patterns (Paudel & Sah 2003; Manral et
al., 2020) as well as biological responses to biotic and abiotic factors (Bargali et al., 2019; Bisht
et al., 2022). A rapid change in climatic conditions, even at short distances, results in
pronounced heterogeneity in soil types and their chemical and physical properties (Bargali et
al., 2018). The effect of drought conditions on the architecture and ecology of different soils





largely depends upon the soil composition, and therefore, the impact of drought on the soil
architecture, and biotic and abiotic composition needs to be understood in the different climatic
regions. Here, we have brought forth some findings on the different cropping systems on
various parameters of the soil in a drought hit region in India to suggest that multiple cropping
systems could help in retaining and /or recovering the vigour of the soil.
The quantifiable measures of physical parameters act as soil health indicators that confirm soil
fertility. One of the main properties of the soil is its texture, which decides the feasibility of the
production of crops. The increase in the percentage of clay in drought–hit soils enhances the
chances of retention of water and nutrients, which seems to favour crop productivity. The
proportion of sand in LWA land is higher than silt and clay compared to other lands with
different crops; this might be due to the addition of sand due to lack or lesser plantations, which
replaces the silt with sand through soil erosion phenomenon (Bargali et al., 2018). This brings
out that fewer crops in a region show the signs of high sand, and less clay soils further decrease
the existence of nutrients. The soil's water holding capacity determines the water retention in
the soils to nurture productivity; the PCM lands have high water retention compared to other
lands; due to multiple cropping, the retention ability has increased, which correlated with
Kirkegaard et al., (2014) findings. Moreover, due to irrigation facilities for an extended period
of time resulted in surface water retention (Zhang et al., 2019) compared to LWA. The PCM
systems with different cropping seem to modify the soil texture by their litterfall, and the action
of microbes probably makes the soils retain water. Cultivated land has shown higher bulk
density than uncultivated land, as continuous cultivation results in the compaction of the soil
layers compared to fallowed land (Markewitz et al., 2002; Bizuhoraho et al., 2018). Rotation
of crops decreases bulk density and increases soil sustainability (Ouda et al., 2018), but in our
study, CP has shown lower bulk density compared to PCM systems and ACS systems, possibly
due to the presence of water close to these crops, which have moistened the subsurface soils
due to drainage. High soil temperature and less moisture content in OL indicate that the lack
of plants exposed the soil to an increase in temperature and loosened its moisture content.
The pH of the soil in the cultivated lands was higher than that of the uncultivated land, which
might be due to the addition of fertilizer interacting with hydrogen ions. During nutrient intake,
plants release $H^+$ ions, which, in turn, absorb nutrients (Bhatla et al., 2018). In the case of PCM
systems, soil pH is low in the surface layers of 0-15 cm, which may be due to the addition of
nitrogen fertilizers in which the ammonium gets converted to nitrate through nitrification,



resulting in the release of $H^+$ ions. The nitrates that got released might combine with cations
like calcium and magnesium, which are likely to leach to the subsurface, leaving surface soils
with low pH, as our studies have correlated to **Gikonyo** et al. 2022.The values of pH differ
from topsoil to subsurface soils; soil carbon and soil nitrogen are negatively correlated to pH,
so lower pH exhibits more accumulation of organic matter. Most plants grow well when soil
contains mineral acids; and if the soil pH is in the range of 6.0 to 7.0 most bacteria act on the
organic matter and release mineral acids. Apart from organic matter pH also determines the
usability of phosphorus, if soil is too acidic then addition of phosphorus reacts with iron or
aluminum instead of intake to plants, and if soils are too alkaline, then it reacts with calcium
and becomes unavailable to soil (Rosen 2014). The electric conductivity of the soil maintains
the exchange of cations between crops and the soil. The cultivated land shows higher soil
electrical conductivity than uncultivated lands, as adding fertilizers enhances EC; our findings
correlated to Adingo et al. (2021). The soil electrical conductivity is positively correlated
(0.033) with nitrogen and (0.065) with phosphorus, but negative correlations were shown with
potassium (-0.054). The OL systems have shown higher EC compared to other cropping
systems, probably due to high evaporation and transpiration from the plants, which might led
to the deposition of salts at the root zone. The total organic carbon (TOC) showed a significant
difference between different agriculture cropping systems, and it has ranged in a manner of
OL>PCM>CP>ACS>LWA from 0.24% (LWA) to 0.9% (OL); this might be due to open lands
were not disturbed for many years, resulting into the natural persistence of TOC compared to
other lands (Batjes 2014). Like TOC, soil organic matter (SOM) plays an important role in
maintaining nutrients; the concentration varies from 0.03% (LWA) to 1.01% (OL). SOM
improves the soil functions of storing and supplying macro and micronutrients and finds one
of the indicators to determine the productivity and management of different cropping patterns
(Sharma et al. 2020). SOM promotes the soil to handle acidity and helps mineral decomposition
in the soil; furthermore, the soil ventilation and decomposition of SOM results in lower carbon
content (Srivastava & Singh 1989). In the present study, the carbon content is 84% in all the
crop-cultivated lands compared to open lands. Less loss of nitrogen has been identified due to
continuous cultivation other than nitrogen, which has shown above 90% of its availability in
all cultivated lands compared to uncultivated land. Due to the continuous supply of fertilizers
to the soil, it retains the presence of nitrogen; similarly, phosphorus also indicated 85% of its
concentration in cultivated land compared to uncultivated one.



The variation in microbial biomass is also shown to be significantly different in different
cropping patterns. The change in the soil environmental patterns might alter the microbial
biomass either on the surface or inner layers of the soil (Wang et al., 2018). Microbial biomass
maintains the chemical cycling and physical properties, as it is considered as sensitive indicator
that favours organic and mineral fertilization. A study by Rice et al. 1997, that microbial
biomass is homeostatic under optimum conditions. If soil lacks nitrogen, carbon or phosphorus
then the limitation of microbial biomass might be noticed, and excess leads to over saturation
of microbes. So, soil microbial biomass is considered to be a sensitive indicator. Throughout
the agricultural cropping systems, $C_{mic}$ has in a series of 134.2±1.2 µg/g to286.6±1.33µg/g for
all seasons, $N_{mic}$ recorded 11.3±1.3 µg/g to69.5±0.98µg/g and $P_{mic}$ has in the range of 07.6±1.5
µg/g to 77.5±0.6µg/g in three seasons. The results clearly indicate that the accumulation of
debris from plants and perennial roots favours the microbes to deposit biomass, enhancing the
soil's C, N, and P (Latati et al., 2017; Kirkegaard et al., 2017). There has been a positive
relationship (p<0.001) between $C_{mic}$ and soil carbon, which indicates that the richness of
organic matter enhances microbial biomass with higher levels (Bargali et al., 2019). PCM has
a diverged litter to the soil, and rotation of crops with multiple crops might enhance the release
of organic carbon to the soil; thereby, increased $C_{mic}$ would have been noticed. The PCM
systems enhance the clay percentage (Table-2), which seems to be favourable for the microbes
to retain longer periods due to the retention of water and organic carbon, which enriches the
$C_{mic}$. Further, $N_{mic}$ has shown higher accumulation where there has been a close connection
with soil carbon, as nitrogen is available in organic form in the soil where heterotrophic bacteria
require carbon as a source of energy. Due to the presence of leguminous crops in PCM systems,
the prevalence of nitrogen-fixing bacteria does survive in the vicinity of the root zone (Xing et
al., 2022), which likely stimulates the $N_{mic}$ in the soils. Similarly, $P_{mic}$ also has a positive
correlation with $C_{mic}$ and plays an important role in microbial function and activity (Zhang et
al., 2019; Wang et al., 2022; Manral et al., 2023). Compared to ACS, CP, LWA, OL, PCM
systems have crop rotation and deposition of different crop residues in the soil, which supplies
organic phosphorus to the soil and gives an impression that encourages the Gram positive
bacteria to convert them into $P_{mic}$. The cropping systems of PCM have a convincing impact on
microbial biomass compared to other systems due to good soil properties and litter diversity.
Different litters of multiple crops might invade different microbes, which releases nutrients to
the soil in a better way, further resulting in enhanced biomass (McDaniel et al., 2014; Padalia
et al., 2022).




Different cropping systems produce various residues and root exudates, which boost microbial
activity and diversity of the soil. Further, increases soil microbial biomass and establishes C
and N cycling (Li et al. 2019). Thus, the PCM systems with different leguminous and non-
leguminous plants have shown better soil microbial biomass when compared to other cropping
systems of the study.
Microbial biomass richness mostly depends on the soil carbon instead of nitrogen, though the
nitrogen might be influenced by the C:N ratio; similarly, the presence of phosphorus also has
an influence on biomass carbon with the C:P ratio. The maintenance of the ratio between C: N:
P may be due to the litter of multiple cropping, which correlates with the findings of Xu et al.
(2013). The $C_{mic}/C$, $N/N_{mic}$, and $P/P_{mic}$ percent ratios were in the range of 1.21±0.87 to
1.64±0.76, 0.89±0.89 to 3.89±1.4, 0.98±1.4 to 2.99±0.9 respectively. Our study report fell into
the 1.2 to 2.7% range per Devi & Yadava et al. (2006). A study by Ravindran & Yang (2015)
found that the ratio of Cmic/C, N/Nmic, and P/Pmic percent was in the range of 1.2 to 3.1% in
the forest soils, which is similar to ours. The PCM systems in the study have significantly
maintained the abiotic characteristics of the soil, which further improved the plant growth in
drought-hit soils.

## 5 Conclusion

The continuous cultivation deprives the soil of its inherent pool of nutrients, and slowly, the
soil becomes unhealthy. To improve the physicochemical and biological characteristics of the
soil needed for better crop productivity, multiple cropping and crop rotation with other
agriculture practices are suggested. In the present study, we employed these practices in the
drought-hit soil of a different regions in the Anantapur district of AP, India. We demonstrated
that the proper root system and perennial crops play a key role in enriching the microbial
biomass through their diverse litter that invades different microbes. These changes in the
microbial community seem to have affected carbon, nitrogen, and phosphorus levels.
Moreover, the increase in the concentration of soil organic matter has enriched the diversity of
the plants to be grown in the drought regions, making a feasible environment to activate the
microbial diversity with different plant communities. Plants growing in these soils would
uptake nutrients and release H+ ions, making soil pH acidic and enhancement in WHC, SM,
and ST most suitable for sustainable crop productivity. However, further studies would be
required for the detailed characterization of microbial diversity in different types of systems
used in the present study. and how do they change in response to multiple cropping systems



under PCM. The available results clearly suggest that the perennial crop selection in the
drought-hit region has significantly improved the total organic carbon, nitrogen, and
phosphorus sources in the soil, further improving the interaction of microbes and thereby
maintaining the microbial biomass of the soil.
**Conflict of Interest**: There is no conflict of Interest.
**Acknowledgements**: The authors are thankful "MURTI" (Multi-disciplinary Unit of Research
on Translation Initiative) and "GSS" (GITAM School of Science) of Gandhi Institute of
Technology and Management for providing support in the conducting field studies, necessary
instrumental facilities in the laboratory.
**Authors contribution**
The author's contribution is conceptualization by MKR, NJS, investigation by NJS
methodology by MKR and NJS, data curation by MKR & NJS, initial writing by NJS, Writing
– writing and editing MKR, NJS, SSB. All authors discussed, reviewed, and agreed to the
publication.

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

489                             Table:1 Cropping System in Drought Region

| Pattern of Crops | Crop species | Pot Herbs |
|---|---|---|
| Open Land (OL) | Mangifera indica, Solanum lycopersicum, Citrus sinensis, Ricinus communis, Citrus linetta | Chrysanthemum indicum, Rosa rubiginosa, Justicia infundibuliformis |
| Annual Crop with single species (ACS) | Zea mays, Sorghum, Oryza sativa, Solanum melongena, Pennisetum glaucum | Tagetes erecta, Chrysanthemum indicum, Coriandrum sativum, |
| Perennial crops with multiple species  (PCM) | Cajanus cajan, Musa acuminata, Murrayakoenigii, Citrullus lanatus | Jasminum sambac, Chrysanthemum indicum, Spinacia oleracea |
| Less water available lands (LWA) | Arachis hypogaea, Sorghum, Capsicum frutescens, Trigonella foenum-graecum, Helianthus annuus | Trigonella foenum-graecum, Amaranthus caudatus, |
| Crops are grown near the ponds (CP) | Justicia infundibuliformis, Tagetes erecta, Psidium guajava, Portulaca oleracea, Gossypium hirsulum, Citrus limetta | Justicia infundibuliformis, Tagetes erecta, Portulaca oleracea |



| Table: 2 Soil Physical properties of different agricultural cropping systems | | | | | | |
|---|---|---|---|---|---|---|
| | | Agricultural Cropping systems | | | | |
| Parameters | Depth (cm) | OL | ACS | PCM | LWA | CP |
| Sand (%) | 0-15 | 43.12±3.33 | 42.66±2.98 | 36.22±2.89 | 44.25±3.09 | 43.24±2.59 |
| | 15-30 | 42.22±3.37 | 40.16±2.81 | 34.55±2.08 | 43.55±3.04 | 42.19±2.53 |
| | 30-45 | 39.67±2.77 | 38.66±2.70 | 35.43±2.83 | 42.92±2.57 | 41.91±3.29 |



| | | | | | | |
|---|---|---|---|---|---|---|
| Silt (%) | 0-15 | 34.12±2.04 | 27.22±2.17 | 28.99±1.73 | 36.24±2.89 | 37.32±2.72 |
| | 15-30 | 32.12±2.24 | 30.56±2.13 | 31.28±1.56 | 37.12±2.59 | 38.87±2.33 |
| | 30-45 | 35.22±2.11 | 29.98±1.79 | 30.98±2.16 | 36.88±2.58 | 37.88±2.65 |
| Clay (%) | 0-15 | 22.76±1.82 | 30.12±1.50 | 34.79±2.43 | 19.51±1.56 | 19.44±1.36 |
| | 15-30 | 25.66±1.53 | 29.32±1.75 | 34.17±2.73 | 19.33±1.35 | 18.94±1.13 |
| | 30-45 | 25.11±1.25 | 31.36±2.50 | 33.59±2.35 | 20.2±1.24 | 20.21±1.61 |
| BD (g/cm$^3$) | 0-15 | 2.01±0.12 | 1.93±0.15 | 1.89±0.09 | 1.77±0.14 | 1.65±0.13 |
| | 15-30 | 2.32±0.16 | 1.98±0.13 | 1.99±0.15 | 1.85±0.11 | 1.72±0.10 |
| | 30-45 | 3.01±0.24 | 2.01±0.16 | 2.10±0.16 | 1.90±0.13 | 1.89±0.13 |
| WHC (%) | 0-15 | 35.66±2.13 | 44.56±3.11 | 51.26±4.10 | 39.76±2.38 | 33.24±2.32 |
| | 15-30 | 33.72±2.69 | 43.23±3.02 | 49.37±3.94 | 37.12±2.59 | 31.98±2.55 |
| | 30-45 | 34.51±2.41 | 40.12±2.80 | 47.23±3.34 | 35.65±2.49 | 29.67±1.48 |
| SM (%) | 0-15 | 4.08±0.28 | 6.92±0.48 | 9.43±0.66 | 5.32±0.37 | 5.99±0.35 |
| | 15-30 | 3.98±0.35 | 4.94±0.39 | 7.13±0.57 | 4.15±0.24 | 4.27±0.29 |
| | 30-45 | 2.89±2.02 | 3.12±0.15 | 3.42±0.26 | 3.23±0.20 | 2.87±0.17 |
| ST | 0-15 | 32.1±2.56 | 30.2±1.81 | 31.2±0.66 | 29.3±1.46 | 30.1±2.40 |
| | 15-30 | 26.1±1.82 | 24.1±1.45 | 25.6±0.76 | 26.1±1.82 | 29.3±2.05 |
| | 30-45 | 25.2±1.76 | 23.4±1.63 | 24.3±0.12 | 25.7±1.79 | 26.1±1.30 |
| Texture | | Loam soil | Loam soil | Loam soil | Loam soil | Loam soil |






Table: 3 Soil Physical properties of different agricultural cropping systems

| Parameter | Depth | Seasons | Agricultural Cropping systems | | | | |
|---|---|---|---|---|---|---|---|
| | | | OL | ACS | PCM | LWA | CP |
| pH | 0-15 | M | 8.62±0.63 | 8.93±0.43 | 7.79±0.38 | 8.22±0.49 | 8.66±0.43 |
| | | W | 8.0±0.4 | 7.99±0.47 | 8.12±0.48 | 7.8±0.44 | 7.98±0.47 |
| | | S | 7.12±0.46 | 7.72±0.38 | 7.39±0.44 | 7.6±0.53 | 7.35±0.580 |
| | 15-30 | M | 8.92±0.53 | 8.84±0.53 | 7.88±0.55 | 8.45±0.76 | 8.66±0.43 |
| | | W | 7.98±0.55 | 7.8±0.46 | 8.0±0.48 | 7.77±0.64 | 7.7±0.381 |
| | | S | 7.3±0.58 | 7.5±0.37 | 7.2±0.58 | 7.4±0.44 | 7.12±0.36 |
| | 30-45 | M | 8.4±0.50 | 8.6±0.52 | 7.7±0.54 | 8.4±0.58 | 8.2±0.41 |
| | | W | 7.7±0.53 | 7.6±0.37 | 7.9±0.67 | 7.5±0.45 | 7.45±038 |
| | | S | 6.5±0.52 | 7.2±0.57 | 7.1±0.49 | 7.35±0.5 | 7.2±0.36 |
| EC | 0-15 | M | 0.38±-0.026 | 0.13±0.061 | 0.26±0.023 | 0.29±0.026 | 0.22±0.01 |
| | | W | 0.26±0.02 | 0.11±0.05 | 0.22±0.0132 | 0.25±0.02 | 0.21±0.0105 |
| | | S | 0.28±0.351 | 0.12±0.007 | 0.2±0.016 | 0.21±0.014 | 0.11±0.008 |
| | 15-30 | M | 0.28±-0.019 | 0.12±0.011 | 0.2±0.018 | 0.22±0.019 | 0.14±0.012 |
| | | W | 0.22±0.01 | 0.11±0.009 | 0.13±0.0064 | 0.20±0.01 | 0.13±0.009 |
| | | S | 0.28±0.01 | 0.10±0.009 | 0.11±0.0066 | 0.19±0.013 | 0.11±0.005 |
| | 30-45 | M | 0.15±-0.01 | 0.14±0.011 | 0.16±0.011 | 0.19±0.015 | 0.19±0.0114 |
| | | W | 0.26±0.02 | 0.13±0.009 | 0.15±0.0105 | 0.14±0.009 | 0.17±0.102 |
| | | S | 0.28±0.025 | 0.11±0.007 | 0.13±0.0091 | 0.12±0.008 | 0.14±0.0112 |
| TOC (%) | 0-15 | M | 0.91±0.43 | 0.45±0.037 | 0.6±0.04 | 0.24±0.016 | 0.6±0.03 |
| | | W | 0.82±0.35 | 0.33±0.026 | 0.49±0.029 | 0.19±0.0133 | 0.59±0.029 |
| | | S | 0.61±0.050 | 0.24±0.016 | 0.45±0.040 | 0.16±0.013 | 0.52±0.028 |
| | 15-30 | M | 0.77±0.06 | 0.32±0.027 | 0.2±0.012 | 0.19±0.05 | 0.56±0.026 |
| | | W | 0.62±0.037 | 0.29±0.02 | 0.19±0.013 | 0.12±0.01 | 0.3±0.029 |
| | | S | 0.52±0.036 | 0.16±0.011 | 0.09±0.006 | 0.10±0.009 | 0.2±0.016 |
| | 30-45 | M | 0.53±0.037 | 0.21±0.0074 | 0.1±0.009 | 0.18±0.014 | 0.6±0.048 |
| | | W | 0.31±0.015 | 0.123±0.010 | 0.11±0.008 | 0.14±0.0113 | 0.59±0.04 |
| | | S | 0.1±0.009 | 0.10±0.006 | 0.08±0.007 | 0.11±0.008 | 0.52±0.03 |
| SOM | 0-15 | M | 1.01±0.080 | 0.22±0.015 | 0.48±0.043 | 0.03±0.002 | 0.48±0.02 |
| | | W | 0.86±0.077 | 0.01±0.0009 | 0.29±0.023 | 0.02±0.001 | 0.46±0.032 |



| Nutrient | Depth | Season | | | | | |
|---|---|---|---|---|---|---|---|
| | 15-30 | S | 0.499±0.044 | 0.13±0.01 | 0.223±0.017 | 0.01±0.0009 | 0.37±0.018 |
| | | M | 0.77±0.061 | 0.12±0.0088 | 0.12±0.009 | 0.10±0.008 | 0.41±0.032 |
| | | W | 0.51±0.040 | 0.21±0.0177 | 0.11±0.008 | 0.09±0.0072 | 0.06±0.004 |
| | 30-45 | S | 0.34±0.020 | 0.10±0.007 | 0.06±0.0048 | 0.07±0.004 | 0.12±0.0108 |
| | | M | 0.36±0.028 | 0.08±0.0072 | 0.1±0.007 | 0.19±0.015 | 0.48±0.023 |
| | | W | 0.16±0.01 | 0.01±0.0009 | 0.09±0.007 | 0.11±0.005 | 0.46±0.034 |
| | | | 0.07±0.006 | 0.013±0.00066 | 0.09±0.006 | 0.026±0.0023 | 0.34±0.015 |
| N (kg/ha) | 0-15 | S | 1354±67.7 | 1323±66.2 | 1316±65.8 | 1216±60.8 | 1322±66.7 |
| | | M | 1331±66.55 | 1201±72.06 | 1177±58.85 | 1173±70.38 | 1296±77.85 |
| | | W | 1206±72.36 | 186±9.3 | 1186±58.85 | 1156±57.8 | 1243±87.01 |
| | 15-30 | S | 856±42.89 | 729±36.45 | 845±59.15 | 677±33.85 | 987±49.3 |
| | | M | 728±36.4 | 623±37.38 | 633±31.65 | 523±26.15 | 865±43.25 |
| | | W | 655±39.3 | 432±30.23 | 524±31.44 | 416±29.12 | 401±20.05 |
| | 30-45 | S | 254±12.7 | 223±13.38 | 216±15.12 | 206±14.42 | 312±15.6 |
| | | M | 231±13.86 | 201±10.05 | 188±13.16 | 194±9.77 | 296±14.8 |
| | | W | 216±10.64 | 186±11.16 | 196±13.72 | 186±4.72 | 286±17.16 |
| P (kg/ha) | 0-15 | S | 19±1.14 | 18±1.08 | 21±1.68 | 10±1.2 | 14±1.3 |
| | | M | 25±1.5 | 26±1.82 | 36±2.88 | 18±1.08 | 26±1.3 |
| | | W | 32±1.92 | 44±4.16 | 57±2.85 | 27±1.35 | 38±2.28 |
| | 15-30 | S | 17±1.20 | 14±2.3 | 17±0.85 | 5±0.3 | 10±0.5 |
| | | M | 18±1.88 | 20±1.2 | 31±2.17 | 8±0.56 | 19±1.33 |
| | | W | 27±1.35 | 39±2.34 | 49±2.45 | 19±0.95 | 29±2.32 |
| | 30-45 | S | 10±0.6 | 11±0.73 | 11±0.55 | 4±0.24 | 11±0.66 |
| | | M | 8±0.72 | 16±0.92 | 16±0.87 | 7±0.49 | 16±0.8 |
| | | W | 12±0.6 | 25±1.76 | 27±1.35 | 17±1.19 | 22±1.1 |
| K (kg/ha) | 0-15 | S | 267±13.35 | 232±11.6 | 432±21.6 | 132±9.24 | 143±8.58 |
| | | M | 302±15.1 | 421±21.05 | 655±32.75 | 144±11.52 | 198±9.9 |
| | | W | 393±19.65 | 540±32.4 | 829±41.45 | 160±12.8 | 255±12.75 |
| | 15-30 | S | 156±13.32 | 136±9.52 | 242±12.1 | 112±8.96 | 111±5.55 |
| | | M | 222±11.1 | 226±15.82 | 402±24.2 | 125±7.5 | 154±9.24 |
| | | W | 304±15.2 | 430±21.51 | 666±33.3 | 103±8.24 | 178±10.68 |
| | 30-45 | M | 107±6.42 | 102±8.19 | 367±22.02 | 107±6.42 | 98±4.9 |





Table: 4 Different agricultural cropping systems and their microbial biomass of carbon, nitrogen and phosphorus

| Agricultural Cropping system | Seasons | $C_{mic}$ (µg/g) | $N_{mic}$ (µg/g) | $P_{mic}$ (µg/g) | $C_{mic}:N_{mic}$ | $C_{mic}:P_{mic}$ | $C_{mic}/C(\%)$ | $N_{mic}/N(\%)$ | $P_{mic}/P(\%)$ |
|---|---|---|---|---|---|---|---|---|---|
| OL | M | 177.6±8.89 | 47.4±2.65 | 36.3±0.32 | 5.4±0.29 | 3.4±0.27 | 1.43±0.99 | 1.76±0.14 | 1.56±0.12 |
|  | W | 156.3±7.77 | 21.2±1.63 | 18.2±0.98 | 8.9±0.56 | 9.7±0.52 | 1.59±0.12 | 0.89±0.078 | 0.98±0.07 |
|  | S | 134.2±6.2 | 9.44±1.2 | 10.9±0.45 | 13.6±1.08 | 15.6±1.2 | 1.31±0.10 | 1.65±0.14 | 1.43±0.10 |
| ACS | M | 233.1±11.3 | 56.5±3.4 | 46.4±1.1 | 3.4±0.23 | 4.4±0.35 | 1.31±0.14 | 2.55±0.20 | 2.32±1.3 |
|  | W | 209.09±10.96 | 29.3±2.3 | 26.9±1.4 | 5.3±0.43 | 6.4±0.51 | 1.64±0.15 | 1.76±0.14 | 1.56±0.10 |
|  | S | 198.4±9.77 | 11.7±1.1 | 07.6±1.5 | 14.7±1.14 | 17.3±1.21 | 1.43±0.7 | 1.54±0.13 | 1.22±0.09 |
| PCM | M | 286.6±17.33 | 69.5±6.25 | 77.5±0.6 | 4.5±0.22 | 6.7±0.47 | 1.22±0.09 | 3.89±0.31 | 2.99±0.20 |
|  | W | 279.8±20.98 | 45.67±3,19 | 55.9±0.3 | 9.7±0.57 | 11.6±0.65 | 1.46±0.13 | 1.78±0.12 | 1.47±0.08 |
|  | S | 240.3±19.2 | 32.3±1.93 | 36.7±0.5 | 17.2±1.37 | 22.1±1.74 | 1.64±0.16 | 2.77±0.19 | 2.05±0.16 |
| LWA | M | 220.7±13.26 | 43.5±3.07 | 53.8±1.2 | 3.2±0.28 | 4.5±0.44 | 1.86±0.12 | 2.12±0.14 | 2.05±0.18 |
|  | W | 198.5±9.21 | 31.9±1.1 | 30.4±0.5 | 8.6±0.68 | 9.7±0.77 | 1.79±0.10 | 1.54±0.09 | 1.41±0.08 |
|  | S | 176.3±10.65 | 19.3±1.91 | 18.2±0.66 | 14.6±0.79 | 13.5±1.03 | 1.45±0.075 | 1.87±0.09 | 1.65±0.13 |
| CP | M | 198.2±13.76 | 44.8±3.51 | 39.8±0.76 | 5.5±0.44 | 6.5±0.48 | 1.32±0.098 | 2.89±0.20 | 2.16±0.17 |
|  | W | 167.3±0.88 | 26.6±1.33 | 21.2±1.3 | 7.1±0.42 | 8.7±0.52 | 1.21±0.087 | 2.03±0.16 | 1.98±0.11 |
|  | S | 145.6±15.05 | 11.3±0.67 | 08.7±0.67 | 11.5±0.69 | 12.6±1.13 | 1.34±0.12 | 1.85±0.12 | 1.45±0.10 |


Table:5 Pearson's corelation coefficient between agricultural systems, seasons, microbial biomass and physicochemical parameters

| | AS | S | $C_{mic}$ | $N_{mic}$ | $P_{mic}$ | pH | EC | TOC | WHC | BD | ST | SM | N | P | K | Sa | Si | C |
|---|---|---|---|---|---|---|---|---|---|---|---|---|---|---|---|---|---|---|
| AS | 1 | | | | | | | | | | | | | | | | | 1 |
| S | 0.001 | 1 | | | | | | | | | | | | | | | | |
| $C_{mic}$ | 0.264 | 0.233 | 1 | | | | | | | | | | | | | | | |





| | Nmic | Pmic | pH | EC | TOC | WHC | BD | ST | SM | N | P | K |
|---|---|---|---|---|---|---|---|---|---|---|---|---|
| Nmic | 1 | | | | | | | | | | | |
| Pmic | 0.199 | 1 | | | | | | | | | | |
| pH | 0.0887 | 0.698** | 1 | | | | | | | | | |
| EC | 0.12 | 0.234 | 0.576** | 1 | | | | | | | | |
| TOC | -0.498* | 0.556 | 0.63 | 0.725 | 1 | | | | | | | |
| WHC | -0.2* | 0.23 | 0.853** | 0.886** | 0.227 | 1 | | | | | | |
| BD | 0.323 | 0.743* | 0.493 | 0.556 | -0.287 | 0.567** | 1 | | | | | |
| ST | 0.033 | 0.789* | 0.255 | 0.266 | 0.568* | 0.346 | 0.160 | 1 | | | | |
| SM | -0.723** | 0.191 | -0.422 | -0.456 | -0.657* | -0.215 | -0.115 | -0.497* | 1 | | | |
| N | -0.002 | -0.889** | 0.665** | 0.834 | -0.014 | 0.556* | 0.335 | 0.210 | -0.489* | 1 | | |
| P | -0.0034 | 0.533* | 0.756** | 0.886 | 0.066 | 0.623 | 0.365 | 0.256 | -0.561* | 0.487 | 1 | |
| K | 0.076 | 0.489** | 0.486 | 0.578 | 0.014 | 0.598* | 0.443 | 0.223 | -0.443 | 0.465 | 0.478 | 1 |





| | | | | | | | | | | | | | | | | | |
|---|---|---|---|---|---|---|---|---|---|---|---|---|---|---|---|---|---|
| **Sa** | 0.298 | 0.000 | -0.654** | -0.587** | -0.445 | -0.0887 | 0.022 | -0.687** | -0.756** | 0.102 | 0.229 | -0.226 | -0.229 | -0.671 | -0.587 | 1 | | |
| **Si** | 0.498* | 0.000 | -0.140 | 0.012 | 0.032 | 0.199 | 0.022 | 0.006 | -0.654** | 0.109 | 0.139 | -0.127 | 0.156 | 0.007 | 0.001 | 0.273 | 1 | |
| **Cl** | -0.534* | 0.000 | 0.345 | 0.186 | 0.266 | -0.136 | 0.014 | 0.265 | 0.628** | 0.112 | 0.221 | 0.207 | -0.015 | 0.267 | 0.245 | -0.564* | -0.875 | 1 |




Table:6 Pearson's correlation of microbial biomass indices.

| | $C_{mic}$ | $N_{mic}$ | $P_{mic}$ | $C_{mic}:N_{mic}:P_{mic}$ | $C_{mic}/C(\%)$ | $N_{mic}/N(\%)$ | $P_{mic}/P(\%)$ |
|---|---|---|---|---|---|---|---|
| $C_{mic}$ | 1 | | | | | | |
| $N_{mic}$ | 0.961** | 1 | | | | | |
| $P_{mic}$ | 0.882 | 0.912** | 1 | | | | |
| $C_{mic}:N_{mic}:P_{mic}$ | -0.663** | -0.792** | -0.801** | 1 | | | |
| $C_{mic}/C(\%)$ | -0.212 | -0.345 | -0.226 | 0.110 | 1 | | |
| $N_{mic}/N(\%)$ | 0.569** | 0.963** | 0.843** | -0.679** | -0.269 | 1 | |
| $P_{mic}/P(\%)$ | 0.412** | 0.864** | 0.765** | 0.115 | -0.439 | -0.643** | 1 |
