# Peer review of "Impact of Cropping Systems on Macronutrient Distribution and Microbial Biomass in"

_EGUsphere, 2024_

## Referee Comment (RC2)

Review: Impact of Cropping Systems on Macronutrient Distribution and Microbial Biomass in Drought Affected Soils.

**General Comments**

I enjoyed reading the manuscript 'Impact of Cropping Systems on Macronutrient Distribution and Microbial Biomass in Drought Affected Soils', studied in Ananthapuram district of Andhra Pradesh, India. It is difficult to find the aims and objectives. I could say, the aim is to find the suitable cropping systems in drought prone soils.

One important concern is the standard of writing of the manuscript. I could say poor structure of sentences through out the manuscript. It requires restructuring the sentences for easy to read, clear and concise of the meaning and keep the bonding among the sentences in a paragraph and also paragraphs to sections.

**Abstract**

The abstract is to be clear and concise aligning with the title of the manuscript. it could be improved keeping in mind the classical structure of a good abstract.

L11-14 introductory sentence - how these lines are linked to the title. Rewrite the concept to link the objective or the gap of the previous research.

L14-16 objectives - I mentioned earlier about the objective. I could say that 'the objective is to compare the macronutrient distribution and microbial biomass in various land-use types i.e. open lands (OL), annual crops with single species (ACS), perennial crops with multiple species (PCM), less water available lands (LWA), and soil near ponds (CP) in drought prone soils'.

L17-19 methods - poor sentence structure of methods. How these methods are linked to reach the objectives.

L-20-21 data analysis – rewrite it. Overuses of the word 'employed' inappropriate sentence.

L-21-31 Results – rearrange and rewrite the results to support the objectives, not only presenting the data.

L-21-23 Results – Is Microbial biomass carbon (Cmi) the main parameter that helps to to test the r hypothesis/ gain the r objectives. If not, present the indicative parameter.

L-24 - Carbon stock – how it influences? Provide data.

L-27-29 Results – is this the main outcome of the r research to gain the objective? Rewrite.

L-30-31- Outcomes – is this repetitive to the previous sentence.

L-31-33 Outcomes – I could not find "sustainable agricultural practices' used in the other sections of the manuscript, except abstract and conclusion. How does the present research contribute to the concept of 'sustainable agricultural practices'?

L-33 Outcomes – The findings of the resent study (diverse cropping systems) could be useful in the drought-prone soils in other regions to gain higher crop productions.

**1. Introduction**

l-36-39 – This sort of sentence structure is used all-around the manuscript. These are not easy to read and understand and free flow of the topic. Please rewrite these sentences to make it concise and clear meaning.

Paragraph structure- make a topic sentence followed by the relevant information.

Research gap – discuss the relevant topics in introduction and narrow down into the research gap. Link the research gap with the objectives of the present study.

L-82-84 - Findings in introduction? Without knowing the results of the research, how will a reader link/ accept the suggestion /the finding. Better, Remove it.

Fig-1: is it relevant with the r research? It was not mentioned anywhere in the text. Fig-1 could be referred in L-80, L-91 or L-101. Place Fig-1 in appropriate location following the text.

**2. Material and Methods**

2.1 Study area and Climatic conditions -shows 5 different land uses.

2.2 Collection of Soil Samples – defines 10 sampling sites.

Fig-1 shows 10 sampling sites at Ananthapuram district of Andhra Pradesh, India. How these 10 sampling sites are related to 5 different land uses. The 10 sampling sites were not discussed anywhere except in Fig-1 and section 2.2.

L-101-104 – a very long sentence. Poor sentence structure.

L-105- why 3 sub-samples?

2.3 Soil Analysis

L-107-112 – which methods were selected for analysing physical parameters and chemical properties?

L-113 – check the tense

**3. Results**
3.1 Physicochemical characteristics

L-121 – is not results. It could be in methods sections (2.4 Statistical Analysis).

L-122-126 – references?

**4. **Discussion**

In discussion, the manuscript contains several concepts, which could be presented in Introduction to find the research gap and link the objectives with the gap.  In discussion, relate the results to establish the objectives. It could be referred or refuted arguments using other references.

L-178-201 could be used in Introduction to link the research gap with the objectives of the present study.

L-211 -286 very long paragraph. These could be break down into several paragraphs, attaining these objectives.

L-248-258 are the introductory concepts, that be presented in Introduction. It is not worthy to new present concept. It could be used as reference for referring or refuting the arguments/ the present results.

L-252 -253 – very poor sentence.

L- 254- 255 – reference??

**5. **Conclusion**

As I understood, it was a comparative study among land uses in drought prone areas.

Rewrite the conclusion that the objectives were achieved. It is not worthy to present several new concepts in conclusion without discussing in results and discussion. For example, root system, perennial crops and or sustainable crop productivity.

L-310-313 could be the last sentence of the manuscript.

**Additional comments**

The manuscript requires a major change/ restructuring  in presenting the results. Keep in mind that the literature reviews will be presented in Introduction, to find a research gap, which could be the aims of manuscript. The aims will be achieved by several objectives. To gain the objectives, the appropriate methods will be followed. The data/ results will be presented to achieve each objectives, finally, the aims of the manuscripts.

Using personal pronoun 'we', 'our' very frequently through the whole manuscripts, which is uncommon in international journal. It requires to edit through the article.

---

## Author Comment (AC2)

**Replies to the specific comments and suggestions that reviewer made on the manuscript**

Thank you so much for your kind and useful comments

| S.No | Comments & Justification |
|------|--------------------------|
| 1. | **Comment**: Some statements are maybe valid for Andhra Pradesh, but not globally.

**Justification**: Thank you for your comment; I do agree that some statements are very well valid for Andhra Pradesh due to its climatic situations and soil conditions. |
| 2. | **Comment**: Line 48: "In the open land use system, the bulk density is higher because of soil compaction." - why, which effects compact the soil? Maybe, it is the heat in summer time.

**Justification**: In open land use systems, soil compaction contributes to higher bulk density. This is primarily due to the loss of natural soil structure, which includes aggregates and pore spaces. Compaction disrupts these structures, causing aggregates to break down under pressure from soil particles, thus reducing pore spaces and increasing bulk density. Additionally, mechanical compaction from activities like foot traffic or animal movement further reduces pore size, leading to increased bulk density. Moreover, high temperatures can exacerbate compaction by drying out surface soils, making them more prone to compaction. |
| 3 | **Comment**: Line 71: "Soil cropping practices enhance soil carbon, nitrogen and phosphorus contents" - this is a nonsense, and much more complicated. Soil carbon gets enhanced from CO2 aerial uptake resulting in the formation of fine roots, if the roots largely remain in the soil after cropping; this is not the case for e.g. onions, potatoes. Soil nitrogen gets enhanced only from the aerial uptake by leguminous crops. There is, however, almost no P-input from the atmosphere (in Europe about 0,3 mg/ha) with rain and dust. But microbes may change the solubility of phosphates in the soil by weathering P-minerals.

**Justification**: Thank you for your valuable comment the study conducted by Bastida et al. (2017), which clearly states that soil cropping practices enhance soil carbon, nitrogen, and phosphorus content, potentially increasing the diversity of soil microbes. It is indeed valid to consider that atmospheric $CO_2$ serves as a source of enhancing soil carbon, while the retention of crop residue and organic amendments further increases soil carbon by adding organic matter. Additionally, as you rightly mentioned, leguminous crops have the ability to fix atmospheric nitrogen, thereby increasing soil nitrogen content. Furthermore, crop rotation and the use of organic amendments can improve soil phosphorus content. Through the addition of fertilizers and organic amendments in conjunction with different crops and their rotation, there is a substantial increase in soil carbon, nitrogen, and phosphorus. |

| | |
|---|---|
| | **Comment** - Soil carbon gets enhanced from CO2 aerial uptake resulting in the formation of fine roots, if the roots largely remain in the soil after cropping; this is not the case for e.g. onions, potatoes

**Justification**: The enhancement of soil carbon through $CO_2$ aerial uptake occurs via photosynthesis, where plants absorb atmospheric carbon dioxide and convert it into organic carbon compounds. This process supports the growth of plant roots, including fine roots, which contribute organic matter to the soil upon decomposition, thus enhancing soil carbon content. However, in the case of certain crops like onions and potatoes, the roots are typically harvested along with the crop and are not left in the soil after cropping. Consequently, the contribution of these crops to soil carbon enhancement through root retention is limited compared to crops where a significant portion of roots remain in the soil after harvesting. Therefore, the contribution of these crops to soil carbon enhancement through root retention is limited compared to crops with a higher proportion of roots remaining in the soil after harvesting.

**Comment**: Soil nitrogen gets enhanced only from the aerial uptake by leguminous crops. There is, however, almost no P-input from the atmosphere (in Europe about 0,3 mg/ha) with rain and dust. But microbes may change the solubility of phosphates in the soil by weathering P-minerals

**Justification:** The comment that soil nitrogen gets enhanced solely through aerial uptake by leguminous crops is inaccurate. Leguminous crops indeed make a substantial contribution to soil nitrogen levels through biological nitrogen fixation, but atmospheric nitrogen deposition, which occurs through various mechanisms such as precipitation and dust settling, also contributes to soil nitrogen; apart from that, nitrogen also enters the soil through organic matter decomposition, fertilizer application, biological processes like mineralization. In the case of phosphorus, direct atmospheric input in the form of phosphorus is indeed minimal. However, soil microbes play a crucial role in modulating the phosphorus dynamics within the soil. These microbes are involved in the weathering of phosphorus minerals, liberating bound phosphorus, and increasing its solubility. Subsequently, the availability of phosphorus for plant uptake is facilitated by microbial action, further increasing the phosphorus levels in soils. |
| 4. | **Comment**: Line 112: Though exchangeable potassium is rather similar, no matter which extract is used, the extractant solution mentioned in Pratt 1965 should be given, because this book might be hardly available to a broad audience.

**Justification**: Thank you for your suggestion, Patt's (1965) procedure involves the extraction of potassium through ammonium acetate as an extracting solution. This procedure provides a standardized method for assessing the exchangeable potassium content in soil, which is important for understanding soil fertility and nutrient management. |
| 5. | **Comment**: Line 134: "..increase in the pH clearly states the enhanced use of synthetic fertilizers" - This is not generally true. K-salts are neutral (KCl, K2SO4); the natural phosphates are usually acidified to yield superphosphate with sulfuric acid, or hyperphosphate with phosphoric acid. If you deliver nitrogen as ammonium, you acidify the root surface, and if you deliver nitrogen as nitrate, you alkalize the root |

surface. If in Andhra Pradesh a pH-increase from fertilization was noted (see also line 174), the wrong products were selected. Unfortunately, the authors did not deliver a market study, which fertilizers had been sold in the region.

**Justification**: The statement that an increase in soil pH necessarily indicates enhanced use of synthetic fertilizers may not be universally acceptable, as various factors influencing soil pH dynamics include the type of fertilizers applied and their specific chemical properties. It has been agreeable that potassium chloride and potassium sulphate are in nature neutral, which might not affect soil pH on the application; these provide without altering the soil's acidity or alkalinity. By the addition of sulphuric acid or phosphoric acid, natural phosphorus is often processed as superphosphate or hyperphosphate, which leads to acidification. However, the degree of acidification depends on the type and amount of phosphate fertilizer used. Similarly, different nitrogen sources can influence soil pH differently; ammonium tends to acidify the root surface upon nitrification as it releases hydrogen ions into the soil, and if it is nitrate-based, then it leads to an increase in pH due to the release of hydroxide ions during nitrification. Now in the current study in Andhra Pradesh, if a pH increase has been noticed resulting from fertilization, which suggested that the selection of fertilizer may not have been suitable for the soil conditions and crop requirement, this could be due to factors such as the availability of fertilizers in the market, farmer preferences or lack of awareness about soil pH management, which was identified.

| 6. | **Comment**: Line 155: "...accumulation of organic carbon in perennial crops ..." which perennial crops were considered (table 1)?, was the straw taken away, or left on-site? This is too much simplified! |
| --- | --- |
| | **Justification**: In the study, the microbial biomass carbon, microbial biomass nitrogen, and microbial biomass phosphorus concentrations vary significantly across all cropping systems, with perennial crops exhibiting higher concentrations compared to others. However, the question arises whether the observed differences in MBC, $N_{mic}$, and $P_{mic}$ can be attributed solely to the presence of perennial crops or other factors also play, there might be such as the fate of crop residues such as straw which influence soil microbial biomass and nutrient dynamics if the straw is removed it can lead to decrease in soil organic carbon and nutrient availability impacting microbial biomass if the straw is left, it can provide carbon and nutrient source for soil microbes, enhancing microbial biomass. Secondly, perennial crops often have a deep root system and more root biomass compared to annual crops, which might increase carbon inputs through root exudation, promoting microbial biomass accumulation, thirdly seasonal variation, and soil management practices improve soil organic matter. Moreover, soil properties also influence microbial biomass and nutrient availability. |
| 7. | **Comment**: Line 193-195: more sand in LWA-land might be explained by the fact that weathering is favoured by microbial activity. The result of weathering of primary silicates are either clay minerals, or pedogenic oxides, which are mainly found in the clay-size fraction. Addition of sand seems obscure - the authors should explain why they had this idea. |

| | |
|---|---|
| | **Justification**: Microbes have a significant role in mineral weathering processes by secreting organic acids and enzymes that break down mineral particles, including sand. While microbial weathering typically affects mineral composition rather than soil texture directly, it can indirectly influence soil texture dynamics. The idea of sand addition in LWA land leading to higher sand proportions compared to silt and clay in other lands with different crops may seem hypothetical; it is grounded in the understanding of soil erosion processes (water erosion, wind erosion that can selectively remove finer soil particles, leaving behind coarser particles like sand) land management practices (where tillage, cultivation, and irrigation can disturb soil structure and promote soil erosion, that can lead to the displacement and redistribution of soil particles, with finer particles like silt and clay being more susceptible to erosion than sand), and soil texture dynamics. |
| 8. | **Comment**: Line 198: " ... the PCM lands have higher water retention ..." - this is logical. Fine roots remain in the soil after removal of the crops, which degrade or are fed by worms, and the resulting channels can be filled with water.

**Justification**: The higher water retention capacity observed in PCM lands compared to other lands can be attributed to a combination of factors, including organic matter accumulation, root development (extensive root systems create channels for water infiltration and extraction from deeper soil layers increases water retention capacity), soil structure and aggregation (continuous cropping increase stability of soil aggregates which creates macropores and micropores that hold water and allow for better water movement), crop rotation (different crops have varying root architecture and water uptake patterns which can complement each other), reduction of soil erosion, and microbial activity (enhancing soil aggregation and cycling nutrients). |
| 9. | **Comment:** Line 204: ".. continuous cultivation results in the compaction of the soil layers .." --> too heavy agricultural equipment? Increased water consumption by crops? Formation of roots should have the opposite effect!

**Justification**: True, continuous cultivation results in the compaction of soil layers, loss of soil structure (mechanical action compacts the soil, reducing pore spaces and increasing bulk density), reduction in organic matter (decline in soil organic matter can contribute to soil compaction and higher bulk density), decreased microporosity (reduction in microporosity limits water infiltration and air exchange further enhances bulk density), and loss of soil aggregation (extensive cultivation can break soil aggregates which reduces the stability of soil structure). |
| 10 | **Comment**: Lines 229/230: for me, this is no correlation!

**Justification**: A coefficient close to zero implies a weak or negligible linear relationship between soil electrical conductivity (EC) and nutrient concentrations. While these correlations may not be practically significant, they still provide valuable perceptions of potential associations between variables. Although the correlation coefficients for nitrogen and phosphorus are small, they indicate a slightly positive relationship between soil EC and the concentrations of these nutrients. This indicates that as soil EC increases, there may also be a slight tendency for nitrogen and phosphorus concentrations to increase. While the correlation is weak, it may still reflect fundamental processes such as nutrient cycling, soil fertility, or management practices that influence both soil EC and nutrient availability. Similarly, the negative |

| | |
|---|---|
| | correlation coefficient for potassium suggests a weak negative relationship between soil EC and potassium concentrations. This means that as soil EC increases, there may be a slight tendency for potassium concentrations to decrease. Again, while the correlation is weak, it may reflect factors such as soil mineralogy, nutrient uptake by plants, or fertilizer applications that influence both soil EC and potassium availability. |
| 11. | **Comment**: Line 254: "..oversaturation of microbes.." what should that be? If there are too many nutrients plus water, there is danger of anoxia, sulfate reduction, ammonia formation and the like!

**Justification**: Yes, nutrient limitation can constrain microbial biomass and activity in soils, while nutrient excesses, combined with waterlogging, can lead to the over-saturation of microbial populations and the potential for detrimental processes such as anoxia, sulfate reduction, and ammonia formation |
| 12 | **Comment**: Line 290: when you consider the P/P-mic ratio, do not forget that the P is from the Olsen extract, which has been invented as exchangeable versus bicarbonate; this is not the total or the CAL-P!

**Justification:** By comparing the P/Pmic ratios obtained using the Olsen-extract method with either total phosphorus or CAL-P, one can assess the relationship between phosphorus availability and microbial biomass phosphorus using different phosphorus extraction methods. This comparison provides comprehension of the dynamics of phosphorus cycling in the soil and its availability to microbial communities. |

---

## Author Comment (AC3)

COMMENTS

**General comments**

This manuscript addresses the impact of cropping systems on macronutrient distribution and microbial biomass in drought affected soils. Soil samples were taken from various land types differing in terms of crops and water availability. Based on the physicochemical characterization and fumigation, the distribution of macronutrients and microbial biomass were assessed in three depth increments down to 45 cm. ANOVA were used to identify differences across cropping systems. Pearson correlation analyses were used to identify relationships between microbial and physicochemical soil parameters. Perennial crops with multiple species show the highest content in microbial biomass, nutrients and water-holding capacity compared to all other crop systems. The authors conclude that diverse cropping systems can effectively enrich soil nutrients and biomass content during drought stress.

**Justification**: Thank you for your comment

While this issue is relevant in terms of the increasing risk of drought to soils and sharing best practice on how to adapt agricultural practices accordingly, the manuscript does not provide novel results or convincing data. In general, the manuscript contains many too general and repetitive statements, and the wording is hard to follow. The manuscripts do not provide any specific research questions, testable hypotheses or theory driven rationale. Thus, there is no storyline which could guide the reader through the manuscript. The method section does not provide details on how the physicochemical and microbial analysis were conducted. Thus, the confidence of this data cannot be evaluated. The discussion is not data-driven and contains too may speculative and over-simplified statements. Since there are so many concerns and flaws regarding the scientific quality (see details below), I am afraid that this manuscript is not suitable for publication.

**Justification:** Thank you for your comment, here are few justifications about the work which was conducted, trying to justify each and every comment.

**Below are the edits and comments on the manuscript**

*Abstract*

Does not provide the novelty of the manuscript.

**Justification**: Thank you for your comment; I believe that the abstract does provide some novelty of the manuscript, as the manuscript underwent assessment of different land types, which included open lands, annual crops, perennial crops and less water available lands, crops close to ponds to elucidate the distribution of macronutrients and microbial biomass. Studying different land types in relation to microbial biomass and soil health provides insights into the diversity of cropping systems, secondly, focus has been made particularly on drought-affected regions where the impact on soil microbial biomass and nutrient utilization remains unexplored, by investigating the response of soil health parameters to drought stress, the study addresses a significant gap in understanding sustainable agricultural practices in such regions. Furthermore, multiple cropping systems highlighted mitigating the impact of drought on macronutrient impact and soil microbial biomass. Also identified that the PCM exhibited superior nutrient availability and microbial biomass.

Line 26 – 33 is highly repetitive.

**Justification** : Thank you for your response from Lines 26 – 33. There are some key points, such as the importance of multiple cropping systems, specifically perennial crops with different species, increasing microbial biomass and nutrient levels in the drought-affected regions. This paragraph also provides information relating to enhancements in soil moisture and macronutrients such nitrogen, phosphorus and potassium which may be may not be considered repetitive. Overall, the whole paragraph does not repeat the main idea, it also introduces new information focusing to a complete understanding of the study findings.

*Introduction*

**Comment:** The intro is too general and hard to follow. The intro does not show in how far this manuscript provides novelty. It does not elaborate on the state of the art properly and how far this manuscript addresses the research gap. The introduction lacks specific research questions and testable hypotheses.

**Justification:** The introduction focuses on the relationship between soil microbial biomass, soil health, agricultural practices, and ecosystem sustainability; it emphasizes how microbial biomass in soil plays an important role in maintaining soil organic matter and biogeochemical cycles. The introduction has shown a glimpse of exploring relationships between cropping patterns, soil microbial biomass, drought resistance, and nutrient cycling, specifically in the Andhra Pradesh region. The state-of-the-art begins with the existing knowledge on the impact of land use patterns, soil properties, and nutrient availability on drought occurrence and soil health; it specifies the role of microbial activity in enriching soil nutrients and demonstrating the influence of cropping practices on soil microbial communities. There are a few questions that were tried to address, such as how different cropping systems influenced soil microbial biomass in drought-prone regions. Dynamics of soil microbial biomass in terms of soil depth and seasonal variation in these regions. The article also tried to understand the relationship between soil microbial biomass and nutrient composition, such as carbon, nitrogen, and phosphorus in drought-affected soils. The novelty of the work represents the influence of multiple cropping systems on soil microbial biomass in drought regions, focusing on the soil depths, seasonal variation, diversity, and nutrient composition.

**Comment:** Line 36: You need to specify what microbial biomass is indicative for.

**Justification:** Thank you for the suggestion of microbial biomass indicative of the activity of microorganisms in the soil, which is helpful in the decomposing of organic matter; biomass includes the consortium of bacteria, fungi archaea, indicative of soil capacity of decomposition, nutrient cycling to assess the quality and fertility of the soil.

**Comment:** Line 39 – 47: This section contains many generic statements and is repetitive. What factors in detail are affecting microbial density and how? Be more specific.

**Justification:** Thank you for putting these in line, as the paragraph does meet on several related points, not necessarily to repeat the same information, instead, it provides a holistic overview of the factors influencing the stability of soil and microbial density, the para informs about the stability and microbial density in different ecosystems and highlighted the impact of agricultural practices on soil health, it connects the colonization of microbes with nutrients and clarifying how the implementation of organic contents which affects soil characteristics and microbial biomass. The lines are reframed as:

"The stability of soil and microbial density is influenced by the type of ecosystem and availability of nutrients (Dietterich et al., 2022; Manral et al., 2023). In most agricultural practices, the widespread use of fertilizers, hybrid seeds, and pesticides contributes to the degradation of the environment, particularly to soil health. Crop productivity heavily relies on

the availability of nutrient levels in soil, which plays an important role in supporting microbial biomass. The application of organic materials enriches the soil with nutrients, helping in the colonization of microbial communities (Bastida et al., 2017). Consequently, changes in soil characteristics may become apparent."

**Comment:** Line 48: What do you mean with open land use systems in that context?

**Justification:** Here, open land refers to a region that went through natural compaction processes typically due to gravitation forces or the trampling of animals, other factors like the settling of soil particles, and heavy machinery moving through the area, which has not been cultivated for a very long time.

**Comment:** Line 52 – 58: Too general. Provide more details and show how this is relevant to your topic.

**Justification:** Thank you for your comment; lines 52 – 58 provide several important aspects of the relationship between vegetation, formation of soil organic matter, soil processes, and productivity of the soil. The paragraph will be modified as

"The vegetation employs a great effect on soil organic matter (SOM) dynamics and basic soil – forming methods, such as aggregation and podozolization (Awasthi et al, 2022 a & b). The organization of plant species within an ecosystem models the quantity and quality of organic contributions to the soil, thus inducing SOM organization. Furthermore, the impact of vegetation expands beyond organic matter inputs. Han et al. (2021) highlighted the role of floristic composition in determining SOM dynamics and original soil-forming processes. Certain plant species may release root exudates that enhance microbial activity and SOM yield, while others may have deeper root systems that influence soil aggregation and structure. Additionally, soil texture interacts with vegetation to influence soil productivity. It has an influence on the moisture retention and availability of nutrients, which further impact the microbial decomposition and cycling processes."

**Comment:** Line 59 – 61: How does drought affect soil erosion?

**Justification:** Thank you for your comment, drought significantly has a major impact on soil erosion, which has interconnection with changes in the land use pattern such as reduced vegetative cover, reduced soil stability as increasing soil compaction making soil less stable, which decreases infiltration and increases surface runoff, which further increases wind erosion where loose dry soil particles get lifted and drifted up.

**Comment:** Line 61 – 65: Hard to follow. Rock-weathering is not the only input of nutrients into the soil. Especially when we talk about cropland systems with fertilizer use.

**Justification:** Thank you for the comment, I have just modified the statement as The supply of nutrients in the soil is influenced by various processes, including rock weathering, decomposition of organic matter, and external inputs such as fertilizers (Smith et al., 2020), while rock weathering contributes minerals to the soil over long time scales, it could be one of the sources of nutrients, along with the use of fertilizers in the agricultural systems. As fertilizers provide the available macronutrients such as nitrogen, phosphorus, and potassium to support plant growth and productivity of crops. These key ingredients are also influenced by microbial activity. Furthermore, the microbial processes contribute to the transformation of nutrients between different chemical forms, making them available for plant uptake.

**Comment:** Line 65 – 67: I do not understand this statement. What do you mean with interaction and physical change in this context?

**Justification:** Thank you for your valuable comment. The statement implies that managing cropping patterns to incorporate organic matter into the soil can enhance the interaction between soil and microbes by promoting microbial activity through both physical changes to the soil environment and nutrient supply. When we talk about, maintaining the interaction between soil and microbes, we refer to managing the conditions that support microbial activity and function of the soil.

**Comment:** Line 70 – 72: Too general. Be more specific.

**Justification:** Thank you for your comment about resorting to soil microbial communities, optimizing nutrient cycling, and improving crop productivity and agricultural practices. Much focus has to be put into reducing drought stress. As per the author, diversified cropping systems support wider microbial taxa and their functional groups, compared to monoculture systems. This will improve the diversity of microbes which can increase the strength of soil ecosystems to abiotic stresses.

**Comment:** Line 75 – 77: This statement is not true. There are already many studies published dealing about drought-affected cropping systems and their nutrient cycling. You need to show in how far your study differs from previous once and what the novelty is.

**Justification:** Thank you for your suggestion; though there were many studies have been conducted in the past, the assessment of soil microbial biomass in relation to different cropping systems in drought-prone regions has been very few. Previously studies may have explored soil microbial biomass in different cropping systems but here our study focussed on drought-hit regions of Andhra Pradesh, as the area experiences water scarcity which adds a unique dimension to the research, secondly in the past the relationship between carbon, nitrogen, and phosphorus may not be extensively explored but our study fills this gap by investigating the influence of multiple cropping systems on soil microbial biomass and nutrient content. Moreover, our study also has also focussed on how different cropping patterns impact soil health in drought-hit regions, which may not have been explored thoroughly in past studies.

*Material and Methods*

The method section is not specific enough. Based on the provided information, it is not possible to evaluate the data quality.

**Comment:** General question: How was seasonality treated in the analysis? How did you differentiate between seasonality effects and drought effects?

**Justification:** The method adopted by collecting samples in three distinct seasons such as summer, monsoon, and winter, allows for examining the potential seasonal variation in soil characteristics and soil microbial biomass; by collecting the samples across the seasons, the study goal is to apprehension any sequential changes in soil properties and microbial communities that may occur in response to seasonal fluctuations in environmental conditions such as temperature, precipitation, and plant growth dynamics.

**Comment:** Line 90: More information needed. Is the study region drought-affected the whole year or during specific seasons?

**Justification:** The study area is located in the southern part of Andhra Pradesh, Anantapur district, predominantly semi-arid climate with distinct wet and dry seasons. The study area

often faces prolonged dry spells and water scarcity, leading to drought conditions; the dry season is mostly found from October to June, showing high water scarcity and low rainfall, and high temperatures. As it is seasonally based, not the entire year was affected by the drought.

**Comment:** Line 93: More information is needed about "less water available lands" and "crops grown near the ponds". This category is based on what parameters? Do they differ in soil moisture, soil temperature, electrical conductivity, etc.?

**Justification:** Less water available lands have limited access to water, making them more prone to drought stress; crops near the ponds have close proximity to ponds and reservoirs, including parameters such as soil moisture, soil temperatures, and electrical conductivity. In case of less water availability, lower soil moisture, high soil temperature, and higher electrical conductivity will be recorded due to increased soil salinity due to water-stressed conditions. In the case of crops near the ponds, increased soil moisture, lower soil temperature, and lower electrical conductivity due to reduced soil salinity.

**Comment:** Line 108: What the soil temperature measured in the field?

**Justification:** Resistance thermometers are used to measure accurate and stable temperatures buried at different soil depths for monitoring.

Section 2.3: You need to provide more details about your methods. The reader cannot reproduce your analysis based on the provided description.

**Justification:** Thank you for your comment. The methods that are provided have followed the standard protocols, which have been given references for each method. Here is the description of the protocols

The texture of the soil was assessed by passing soil through a sequence of sieves with varying aperture sizes: Sand (0.02-2.0 mm), silt (0.002-0.02mm), and clay (<0.002 mm). The soil particles composition was then determined by weight, following the method outlined by Misra (1968). Bulk density has been determined using a specialized metal core sampling cylinder with a known volume. The soil moisture content has been calculated gravimetrically by subjecting soils to drying until reaching a constant weight, then expressing the water content as a percentage of the dry weight. The pH levels were measured in a 1:5 mixture of soil and distilled water using a glass electrode. The soil carbon content was measured using the rapid titration method devised by Walkley and Black (1934). Total soil nitrogen (Peach & Tracy 1956) was determined using the micro-Kjeldahl digestion technique, while total phosphorus was measured using a spectrophotometer, following the procedure described by Misra (Olsen et al 1954).

The microbial biomass content has been determined using the chloroform fumigation and extraction method following the protocol modified by Brookes et al., 1985. Ten grams of three portions of moist soil samples were weighed, and two portions were placed in a crucible alongside a shallow dish containing 30 ml alcohol-free chloroform. The other portion was left unfumigated and also placed in a crucible within a separate desiccator without chloroform. Both of them were covered and kept in the dark at room temperature for days Vance et al., 1987. After the fumigant was removed, the fumigated soils were extracted with 0.5 M potassium sulfate; the same was processed with non-fumigated soils. For further analysis soil extract (8 ml), 0.2 M potassium chromate (2ml), concentrated sulphuric acid (10ml), and 85% phosphoric acid (5 ml), were thoroughly mixed and the mixture was then digested at 150 C for 30 minutes and treated 0.1 N ferrous (II) ammonium sulphate, employing 2-3 drops of ferroin indicator.

**Comment:** Line 118: What exactly where you trying to find out by using Pearson correlations? Because table 6 contains autocorrelations.

**Justification:** The use of Pearson correlations has been done to explore the relationship between different microbial biomass such as C ($C_{mic}$), N ($N_{mic}$), P ($P_{mic}$). High positive correlations between these indices would indicate the coordinated response of microbial communities to environmental conditions. Secondly, relative proportions between these ratios provide insights into microbial community composition and pattern of nutrient utilization.

*Results*

**Comment:** The data is poorly presented. The result section already contains interpretations and generic statements.

**Justification:** The results that are provided are well organized and formulated in different tables and most of them are based on statistical analysis of the data, it also highlighted how interpretations contribute to the understanding of the results.

**Comment:** General question: Where are the ANOVA results presented?

**Justification:** ANOVA results are presented in tables-2,3 and 4

**Comment:** Line 122: Similar texture does not imply the same parental rock. And how can you be sure, that in your (sub)tropical soils the sand fraction is not influenced by microaggregates and pedogenic oxides?

**Justification:** Soil texture provides valuable insights into soil genesis and processes, which includes the composition of the parent material, weathering processes, and soil formation mechanisms; while similar textures may indicate some degree of geological similarity, in some cases might not conform with the same parental rock. In the case of microaggregates and pedogenic oxides, it may alter the distribution and composition of soil particles, including a fraction of sand, aggregation, and mineral weathering.

**Comment:** Line 123: Too general. And why is this relevant for your topic?

**Justification:** The present study area is composed of ancient volcanic rocks, which, over time, contribute to the formation of soil, so similar soil texture might be found across the agricultural systems deriving common geological origin.

**Comment:** Line 132 – 133: This is trivial.

**Justification:** Thank you for your comment; the above lines are not trivial, as they emphasize the variation across different agricultural systems, which has significant implications for crop production. The statement highlights the diversity of soil pH conditions within the study areas, variation in the pH may influence crop productivity.

**Comment:** Line 134: Increase in pH can have several reasons and not just fertilizer input.

**Justification:** Thank you for your comment, I do agree that the increase in pH can be for several reasons, but in the case of the study area, enhanced use of synthetic fertilizers and

pesticides to improve crop production. Due to the presence of calcium carbonate deposits naturally, most soils in this region are alkaline.

**Comment:** Line 137: Trivial

**Justification:** Thank you for your comment, but the line 137 is not trivial, as variation in electrical conductivity has an influence on the distribution of ions in the soil. The distribution of ions is crucial for optimizing soil fertility management practices to support plant growth and crop yield.

**Comment:** Line 145: What do you mean with 90 %. Where is this number coming from?

**Justification:** Thank you for your comment; the line 145 represents the total nitrogen in the soil compared to other components.

**Comment:** Line 152 – 167: I cannot follow this section. This is already interpretation and is not linked to any hypothesis. The different season were not described in the method section before.

**Justification:** Thank you for your comment on the lines 157 -162. In these lines, the interpretations aimed to provide an understanding of the observed variation in microbial biomass and nutrient concentrations across different cropping systems and seasons. As interpretations are valuable for insights into the implications of the findings, which are supported by clear hypotheses. The samples were collected during three seasons to recite the importance of considering seasonal effects on soil properties and microbial dynamics.

**Comment:** Line 170: This is simply an autocorrelation.

**Justification:** Thank you for the comment, the use of Pearson correlations has been done to explore the relationship between different microbial biomass such as C ($C_{mic}$), N ($N_{mic}$), P ($P_{mic}$). High positive correlations between these indices would indicate the coordinated response of microbial communities to environmental conditions.

*Discussion*

**Comment:** Since no research question nor hypothesis were formulated before, the discussion part is not linked to any central theme. It contains to many generic and oversimplified statements and speculative parts. Further, the discussion is not data-driven.

**Justification:** Thank you for your comment; we have tried to address the research question of how different land types and cropping systems, particularly perennial crops with multiple species, influence soil microbial biomass and macronutrient distribution in affected regions. The study also tried to address how to various land types such as open lands, annual crops with single species, perennials with multiple species, less water available lands, and soil near ponds impact soil microbial biomass and macronutrient distribution. The study also tried to address the microbial biomass carbon, nitrogen, and phosphorus levels across different seasons and soil depths in each land type. We tried to understand how multiple cropping has an impact on soil moisture, nitrogen, and phosphorus levels in drought-stressed environments. Finally, we tried to address how these findings contribute to understanding sustainable agricultural practices and soil health resilience in drought-prone regions.

**Comment:** Line 178 – 188: Too general. No link to data.

**Justification:** Thank you for your comment; lines 178 to 188 give the importance of soil ecology, nutrient richness, and the impact of drought on soil properties, which has been demonstrated in the results segment, the findings of soil parameters, cropping systems, and their impact on soil microbial biomass has been clearly mentioned.

**Comment:** Line 189 – 191: Trivial. Texture is not the only factor controlling soil productivity.

**Justification:** Thank you for your comment; soil texture is indeed a crucial factor influencing soil productivity; it is one of the many factors that contribute to overall soil fertility and crop production. Other factors also contribute to soil health, such as organic matter content, nutrient availability, pH levels, water retention capacity, as well as biological activity. Consider a wide range of physical, chemical, and biological properties for a holistic approach to soil health assessment. While soil texture provides valuable insights into soil characteristics, it should be interpreted in conjunction with other parameters to obtain a comprehensive understanding of soil fertility and its implications for crop production. Soil productivity can vary significantly across different soil types, even within the same texture class.

**Comment:** Line 192 –197: I do not understand the sentence.

**Justification:** Thank you for your comment; here is the explanation: In LWA lands, the proportion of sand is higher than silt and clay compared to other lands with different crops. This might be due to the erosion of finer soil particles such as silt and clay due to lower vegetation cover and activities of agriculture in LWA areas. As limited crops are present in such regions, there will be reduced soil cover and root structure to protect against erosion, leading to the displacement of finer particles by sand. So, soils in LWA have a higher content of sand than silt and clay.

**Comment:** Line 197 – 198: This is trivial.

**Justification:** Thank you for your comment; the line 197-198 highlight an essential aspect of soil science and agricultural productivity. It represents a fundamental principle in soil science: soil water holding capacity influences water retention, which has a great impact on agricultural productivity.

**Comment:** Line 199 – 200: You are jumping to conclusions. There are many more factors other than crop types affecting water holding capacity (texture, bulk density for example).

**Justification:** Thank you for your comment; there could be many factors influencing water holding capacity, such as soil texture, bulk density, and organic matter content, but multiple cropping may contribute to increased water retention. As the study area, falls into a drought-hit region, multiple cropping may enhance soil organic matter content and improve soil structure, leading to increased water retention, its effectiveness may vary.

**Comment:** Line 200: The existence of irrigation facilities on your test sites were not mentioned in the methods before. This comes as a surprise now. How do you isolate this effect from seasonality effects for example?

**Justification:** Thank you for your comment, irrigation practices can both influence soil properties and microbial biomass, the data also represented that different cropping patterns influence of soil properties and microbial biomass

**Comment:** Line 204: Again, there are more factors influencing bulk density. This is too simplified.

**Justification:** Thank you for your comment; line 204 explains that during cultivation, soil gets compacted, resulting in higher bulk density as one of the several factors that can affect soil density. Secondly, other factors such as soil moisture levels, topography, and land management practices are also influenced. Moreover, soil, climate, and cropping patterns also result in soil nature.

**Comment:** Line 211 –215: This is too general. Why is this relevant for your topic?

**Justification:** Thank you for your comment, lines 211 to 215 highlight the relationship between soil pH, cultivation practices, and nutrient management in drought-hit regions, where maintaining soil health and productivity is essential.

**Comment:** Rest of discussion contains many generic and oversimplified statements and is hard to follow due to wording. The impact of drought stress was not properly discussed besides being a main topic of the manuscript.

**Justification:** Thank you for the comment; the discussion has been focussed on the soil physicochemical parameters of different cropping patterns in drought-hit regions of Andhra Pradesh, where these kinds of study have been low; we tried to find out if the soil microbial biomass has any relevance to the cropping patterns, which was found in case PCM cropping systems. The study also visualized the seasonal variation and its influence on soil

*Conclusion*

**Comment:** The conclusion does not show what new knowledge we have learned from this study.

**Justification:** Thank you for your comment; the conclusions have been focussed on the importance of employing multiple cropping practices to improve the physicochemical and biological properties of drought-hit soils; it highlighted that diverse litter provides a favorable environment for microbial diversity, indicating that the selection of proper root systems and perennial crops can positively influence soil microbial communities. It emphasizes how perennial crops may influence soil pH, water-holding capacity, soil temperature, etc.

**Comment:** Line 298 – 299: Oversimplified

**Justification:** Thank you for your comment; indeed, the statement is somewhat simplified, but it effectively captures a common phenomenon in agriculture: continuous cultivation can indeed deplete soil nutrients over time, leading to soil degradation and reduced fertility.

**Comment:** Line 306 – 312: Plant diversity was not discussed before. This is a new aspect brought up in the conclusion.

**Justification:** Thank you for your comment; the meaning of lines 306-312 was that if the drought-hit regions were improvised with multiple cropping systems, then there could be more chances for diverse plant species to be grown.

---

## Author Comment (AC4)

COMMENTS – Referee – 2

**General Comments**

I enjoyed reading the manuscript 'Impact of Cropping Systems on Macronutrient Distribution and Microbial Biomass in Drought Affected Soils', studied in Ananthapuram district of Andhra Pradesh, India. It is difficult to find the aims and objectives. I could say, the aim is to find the suitable cropping systems in drought prone soils.

One important concern is the standard of writing of the manuscript. I could say poor structure of sentences through out the manuscript. It requires restructuring the sentences for easy to read, clear and concise of the meaning and keep the bonding among the sentences in a paragraph and also paragraphs to sections.

Thank you for your general comments, I tried best to give the justification in all aspects.

**Abstract**

The abstract is to be clear and concise aligning with the title of the manuscript. it could be improved keeping in mind the classical structure of a good abstract.

**Comment:** L11-14 introductory sentence - how these lines are linked to the title. Rewrite the concept to link the objective or the gap of the previous research.

**Justification**: Thank you for your comment, comprehension of the elaborate relationship between water availability, soil nutrients, and microbial biomass is essential for improving plant growth and confirming soil health. Although surface microflora traditionally facilitates mineralization and nutrient cycling, the effects of drought on soil microbial biomass and nutrient utilization have yet to be fully investigated.

**Comment:** L14-16 objectives - I mentioned earlier about the objective. I could say that 'the objective is to compare the macronutrient distribution and microbial biomass in various land-use types i.e. open lands (OL), annual crops with single species (ACS), perennial crops with multiple species (PCM), less water available lands (LWA), and soil near ponds (CP) in drought prone soils'.

**Justification:** Thank you for your comment

**Comment:** L17-19 methods - poor sentence structure of methods. How these methods are linked to reach the objectives.

**Justification:** Thank you for the comment. Soil samples collected from different land types indicate that different land types allow for comparison and analysis of how land use practices influence macronutrient distribution and microbial biomass. The samples were air dried, indicating uniform processing and analysis across all samples because dying prevents microbial activity without altering the composition of the sample, and for a comprehensive analysis, the samples should be subjected to physical, chemical, and biological analysis. The overall methods adopted would give a thorough investigation of the impact of land use types on macronutrient distribution and microbial biomass in drought-prone soils.

**Comment:** L-20-21 data analysis – rewrite it. Overuses of the word 'employed' inappropriate sentence.

**Justification** Thank you for the comment; it can be changed to " Statistical analysis, including ANOVA and Pearson Coefficient, were utilized to discern patterns across seasons, soil depts and microbial biomass.

**Comment:** L-21-31 Results – rearrange and rewrite the results to support the objectives, not only presenting the data.

**Justification:** Thank you for the comment; here is the rewritten version Statistical analyses, encompassing ANOVA and Pearson Coefficient, were utilized to determine patterns throughout seasons, soil depths, and microbial biomass. Microbial biomass carbon (Cmic) showed a range of 134.2±1.2µg/g to 286.6±1.33µg/g, while nitrogen (Nmic) and phosphorus (Pmic) explained variability from 11.3±1.3µg/g to 69.5±0.98µg/g and 07.6±1.5µg/g to 77.5±0.6µg/g, respectively, across all seasons. Furthermore, carbon stock in the upper soil surface positively affected nitrogen and phosphorus maintenance. Remarkably, perennial crops with multiple species (PCM) showed superior Cmic, Nmic, Pmic, and water-holding capacity compared to open lands (OL), less water available lands (LWA), and annual crops with single species (ACS). These findings emphasize the impact of diverse cropping systems, especially PCM, in improving microbial biomass and nutrient levels in drought-affected regions. The observed improvements in soil moisture, nitrogen, phosphorous, and potassium levels suggest that varied cropping systems can effectively enrich soil nutrients and biomass content under drought stress. In conclusion, our study highlights the potential of perennial crops with multiple species in mitigating the impact of drought on soil microbial biomass and macronutrient distribution across different land-use types in drought-prone soils

**Comment:** L-21-23 Results – Is Microbial biomass carbon (Cmi) the main parameter that helps to to test the r hypothesis/ gain the r objectives. If not, present the indicative parameter.

**Justification:** Thank you for your comment. One of the important parameters in assessing the impact of different cropping systems on microbial biomass and macronutrient distribution in drought-affected soils is that it may not necessarily be the sole parameter, but $C_{mic}$, $N_{mic}$, and $P_{mic}$ also play a major role in nutrient uptake. Multiple parameters allow for a more comprehensive assessment of soil functioning and the impact of cropping systems on soil health. Along with soil microbial biomass carbon, $N_{mic}$, and $P_{mic}$ will help in understanding soil functioning and cropping systems

**Comment:** L-24 - Carbon stock – how it influences? Provide data.

**Justification:** Thank you for your comment, the upper layers of carbon stock has a positive influence on nitrogen and phosphorus retention in drought affected soils, thereby contributing to the objective of assessing the impact of different cropping systems on nutrient distribution and microbial biomass.

**Comment:** L-27-29 Results – is this the main outcome of the r research to gain the objective? Rewrite.

**Justification:** The conclusions indicate that diverse cropping systems have a knowing influence on soil nutrient levels and biomass content under drought stress. This highlights the potential of varied agricultural practices to enhance soil health and productivity in drought-affected environments, aligning with the objective of evaluating the impact of different cropping systems on microbial biomass and macronutrient distribution.

**Comment:** L-30-31- Outcomes – is this repetitive to the previous sentence.

**Justification:** Thank you for your comment. The lines 27 to 31 can be rewritten as These findings underline the potential of diverse cropping systems, particularly perennial crops with multiple species, to mitigate the impact of drought on soil microbial biomass and macronutrient distribution. This contributes to our understanding of sustainable agricultural practices in drought-prone regions and highlights the importance of applying such systems to increase soil health and resilience.

**Comment:** L-31-33 Outcomes – I could not find "sustainable agricultural practices' used in the other sections of the manuscript, except abstract and conclusion. How does the present research contribute to the concept of 'sustainable agricultural practices' ?

**Justification:** Thank you for your comment, here is the justification. The present study contributes to sustainable agricultural practices by emphasizing the effectiveness of perennial cropping with multiple species in mitigating the impact of drought on soil microbial biomass and macronutrient supply. Perennial crops with diverse species enhance soil health and resilience by promoting biodiversity, improving soil structure, and reducing dependence on external inputs like fertilizers. This approach forwards long-term soil fertility, water retention, and ecosystem stability, aligning with sustainable agriculture principles. By determining the benefits of such cropping systems in drought-prone regions, the study offers practical insights for promoting sustainable land management practices that balance productivity with environmental conservation.

**Comment:** L-33 Outcomes – The findings of the resent study (diverse cropping systems) could be useful in the drought-prone soils in other regions to gain higher crop productions.

**Justification:** Thank you for your comment; yes, we believe in the study that has been conducted.

1. **Introduction**

**Comment:** l-36-39 – This sort of sentence structure is used all-around the manuscript. These are not easy to read and understand and free flow of the topic. Please rewrite these sentences to make it concise and clear meaning.

**Justification:** Thank you for your comment, the line 36-39 emphasizes the importance of microbial biomass as a key indicator of soil health. Microbial biomass is essential for maintaining organic content in the soil through the decomposition of organic matter. This process is vital for controlling nutrient cycling and sustaining biogeochemical processes in various ecosystems. In essence, the line-36-39 highlights how microbial biomass influences the fertility and functionality of soils by regulating the breakdown of organic materials and facilitating nutrient availability for plants and other organisms

**Comment:** Paragraph structure- make a topic sentence followed by the relevant information.

**Justification:** Thank you for your comment, the introduction outlines several important aspects related to microbial biomass, soil health, and the impact of agricultural practices on soil fertility in drought-affected regions. It discusses the essential role of microbial biomass in maintaining organic content, controlling nutrient cycles, and sustaining biogeochemical processes in ecosystems.

**Comment:** Research gap – discuss the relevant topics in introduction and narrow down into the research gap. Link the research gap with the objectives of the present study.

**Justification:** Thank you for your comment; here in this study, we have found there is a lack of studies examining the microbial biomass in different cropping systems in drought-hit regions and their relationship with soil nutrients. Though the soil microbial biomass and nutrients are important in maintaining soil health, there is a lack of research focusing on the aspect in the study area that is Andhra Pradesh, so there is a need to investigate how different cropping systems influence soil microbial biomass and nutrient levels in drought-hit soils.

**Comment:** L-82-84 - Findings in introduction? Without knowing the results of the research, how will a reader link/ accept the suggestion /the finding. Better, Remove it.

**Justification**: Thank you for your comment, but lines 82-84 describe an overview of the study objectives and scope. Here is the modified version: A study has been taken to investigate the impacts of different cropping systems on soil microbial biomass in drought-affected regions. It is aimed to explore various factors, which include soil depths, seasonal variation, and nutrient composition. By understanding these aspects, the study sought to enhance our understanding of how different cropping systems influence soil microbial communities in environments prone to drought stress.

**Comment:** Fig-1: is it relevant with the r research? It was not mentioned anywhere in the text. Fig-1 could be referred in L-80, L-91 or L-101. Place Fig-1 in appropriate location following the text.

**Justification:** Thank you for your comment; the figure has been placed in line 80.

1. **Material and Methods**

**Comment:** 2.1 Study area and Climatic conditions -shows 5 different land uses.

**Justification:** Thank you for your comment; yes, the study area comprises it.

**Comment:** 2.2 Collection of Soil Samples – defines 10 sampling sites.

**Justification:** Thank you for your comment; yes it is true

**Comment:** Fig-1 shows 10 sampling sites at Ananthapuram district of Andhra Pradesh, India. How these 10 sampling sites are related to 5 different land uses. The 10 sampling sites were not discussed anywhere except in Fig-1 and section 2.2.

**Justification:** Thank you for your comment; these 10 sampling sites were related to five different land use systems; from each system, two sites were picked for the study.

**Comment:** L-101-104 – a very long sentence. Poor sentence structure.

**Justification:** Thank you for your comment, it has been restructured in this way: Soil samples were randomly collected from ten distinct regions at the study site across three seasons: summer, monsoon, and winter. Samples were obtained from varying soil depths, including the upper surface (0-15 cm), subsurface (15-30 cm), and deeper layers (30-45 cm). Upon collection, soil samples were placed in Ziplock bags and transported to the laboratory for further analysis. Subsequently, the samples were air-dried and divided into three sub-samples for subsequent analysis of various soil characteristics.

**Comment:** L-105- why 3 sub-samples?

**Justification:** Thank you for your comment: Three sub samples indicate, triplicates of the sample collected.

2.3 Soil Analysis

**Comment:** L-107-112 – which methods were selected for analysing physical parameters and chemical properties?

**Justification:** The soil particle composition was then determined by weight, following the method outlined by Misra (1968). Bulk density has been determined using a specialized metal core sampling cylinder with a known volume. The soil moisture content has been calculated gravimetrically by subjecting soils to drying until reaching a constant weight, then expressing the water content as a percentage of the dry weight.

Comment: L-113 – check the tense

Justification: Thank you for your comment, the statement has been reframed as soil microbial analysis is estimated by taking the surface soils, as the activity of microbes is expected to be higher in the surface soils.

1. **Results**

    o   Physicochemical characteristics

**Comment:** L-121 – is not results. It could be in methods sections (2.4 Statistical Analysis).

**Justification**: Thank you for your comment. Line 121 gives information about the physicochemical analysis of results.

**Comment:** L-122-126 – references?

**Justification:** Thank you for your comment. Soil texture provides valuable insights into soil genesis and processes, which includes the composition of the parent material, weathering processes, and soil formation mechanisms; while similar textures may indicate some degree of geological similarity, in some cases might not conform with the same parental rock. In the case of microaggregates and pedogenic oxides, it may alter the distribution and composition of soil particles, including a fraction of sand, aggregation, and mineral weathering.

1. **Discussion**

In discussion, the manuscript contains several concepts, which could be presented in Introduction to find the research gap and link the objectives with the gap.  In discussion, relate the results to establish the objectives. It could be referred or refuted arguments using other references.

**Comment:** L-178-201 could be used in Introduction to link the research gap with the objectives of the present study.

Justification: Thank you for your comment; lines 178 to 201 give the importance of soil ecology, nutrient richness, and the impact of drought on soil properties, which has been demonstrated in the results segment, the findings of soil parameters, cropping systems, and their impact on soil microbial biomass has been clearly mentioned, these were also been mentioned in the introduction.

**Comment:** L-211 -286 very long paragraph. These could be break down into several paragraphs, attaining these objectives.

**Justification:** Thank you for the suggestion, it will be well taken

**Comment:** L-248-258 are the introductory concepts, that be presented in Introduction. It is not worthy to new present concept. It could be used as reference for referring or refuting the arguments/ the present results.

Justification: Thank you for your comment; I have just reframed the statement to The significant variation observed in microbial biomass across different cropping patterns underlines the understanding of soil microbial communities to environmental changes induced by agricultural practices (Wang et al., 2018). These changes can affect microbial biomass levels not only in surface soils but also in deeper soil layers, reflecting the intricate relationship between soil environmental patterns and microbial dynamics. The microbial biomass plays a critical role in maintaining the chemical cycling and physical properties of soil, serving as a sensitive indicator of soil health and fertility (Rice et al., 1997). Optimal conditions promote microbial biomass homeostasis; however, deficiencies or excesses of essential nutrients like nitrogen, carbon, or phosphorus can disrupt this balance, leading to noticeable limitations or over-saturation of microbial populations. The recorded ranges of $C_{mic}$ has in a series of 134.2±1.2 μg/g to286.6±1.33μg/g for all seasons, $N_{mic}$ recorded 11.3±1.3 μg/g to69.5±0.98μg/g and $P_{mic}$ has in the range of 07.6±1.5 μg/g to 77.5±0.6μg/g in three seasons and cropping systems highlight the dynamic nature of soil microbial biomass in response to agricultural practices.

Comment: L-252 -253 – very poor sentence.

Justification: Thank you for your comment, and it is rewritten as Optimal conditions promote microbial biomass homeostasis; however, deficiencies or excesses of essential nutrients like nitrogen, carbon, or phosphorus can disrupt this balance, leading to noticeable limitations or over-saturation of microbial populations.

**Comment:** L- 254- 255 – reference??

**Justification:** Thank you for your comment. It is the observation of Rice et al 1997 which has been given in the reference.

1. **Conclusion**

As I understood, it was a comparative study among land uses in drought prone areas.

Rewrite the conclusion that the objectives were achieved. It is not worthy to present several new concepts in conclusion without discussing in results and discussion. For example, root system, perennial crops and or sustainable crop productivity.

**Justification:** Thank you for your comment, in the study perennial crop with multiple species showed a significant availability of nutrients and soil microbial biomass, so it has been mentioned as the root system of perennial crops could be a solution for drought hit soils.

**Comment**: L-310-313 could be the last sentence of the manuscript.

**Justification**: Thank you for your comment. It is well taken, but lines 313 to 316 have given how the selection of PCMs could be effective in drought-hit soils.

**Additional comments**

The manuscript requires a major change/ restructuring  in presenting the results. Keep in mind that the literature reviews will be presented in Introduction, to find a research gap, which could be the aims of manuscript. The aims will be achieved by several objectives. To gain the objectives, the appropriate methods will be followed. The data/ results will be presented to achieve each objectives, finally, the aims of the manuscripts.

Using personal pronoun 'we', 'our' very frequently through the whole manuscripts, which is uncommon in international journal. It requires to edit through the article.

**Justification:** Thank you for your comment, well taken.

---

## Author Comment (AC5)

Referee: 3

General Comments and Justifications

While the manuscript delves into an intriguing topic by evaluating soil properties, particularly biological ones, it exhibits significant weaknesses that warrant addressing prior to publication. Some of these shortcomings have already been highlighted by colleagues, and additional concerns are listed below:

**Justification:** Thank you for your comment. It will be well taken and incorporated.

**Introduction:** This section would benefit from enhancement and expansion to incorporate specific issues pertinent to the studied area. This entails not only referencing cropping systems but also addressing aspects such as water availability, seasonality, cropping practices, etc.

- Line 59: The explanation of the effects of drought and land use patterns is unclear. The text should clarify these patterns.

  **Justification:** Thank you for your comment, In addition to the justification to the previous referees, it can also include that the occurrence of drought can indeed influenced by changes in land use patterns, which can alter the physical, chemical, and biological properties of soil. Changes in these patterns might increase erosion, lead to compaction, disrupt the regulation of water, and cause loss of soil stabilization, which contributes to the occurrence and severity of drought conditions.

- Lines 78-87: It's unclear whether this information is mentioned in the text. Consider relocating it to the Materials and Methods section.

  **Justification:** Thank you for your comment. It draws the primary objective and focus of the study, providing readers with a clear understanding of the research scope and meaning right from the beginning. By maintaining this statement in the introduction, we ensure that readers are immediately informed about the key aim of our investigation and the specific context in which it is conducted.

**Materials and Methods:** This section requires more detailed information about the studied area, encompassing climate/weather, soil type, distribution of cropping systems across sampled regions, sampling methodology and year, etc. Additionally, management practices in cropping systems should be elaborated upon.

- How can LWA and CP be compared or utilized as controls? What is the range of precipitation across seasons?

  **Justification:** Thank you for your comment here. Only CP has been considered as a control because it provides a baseline for comparison against other land use types due to consistent water availability having similar environmental conditions such as temperature and soil moisture, which can help minimize confounding variables when comparing microbial biomass and nutrient distribution, soils near ponds may exhibit relatively stable characteristics due to presence of water, crops growing near ponds may present more natural or undisturbed ecosystems.

- The statistical analysis needs further explanation. Did the authors verify the normality of the dataset before conducting ANOVA? What is the significance threshold? Was seasonality the only factor analyzed within cropping systems, or was it also assessed among regions?

**Justification:** Thank you for your comment; yes, the dataset was verified before being set to ANOVA analysis, and further seasonality and variation among the different cropping systems were assessed.

**Results:** The results are not thoroughly explored. Consider presenting tables and incorporating statistical findings into figures and/or tables.

**Justification:** Thank you for your comment and well taken

**Discussion:** This section is challenging to follow. Providing more contextual details about the study area could enhance readability and comprehension of certain assertions (e.g., L194).

**Justification:** You comment on this is well taken and thank you for your comment; here is the explanation: In LWA lands, the proportion of sand is higher than silt and clay compared to other lands with different crops. This might be due to the erosion of finer soil particles such as silt and clay due to lower vegetation cover and activities of agriculture in LWA areas. As limited crops are present in such regions, there will be reduced soil cover and root structure to protect against erosion, leading to the displacement of finer particles by sand. So, soils in LWA have a higher content of sand than silt and clay

**Conclusions:**

- L302-304: The claim here lacks substantiation.

  **Justification:** Perennial crops with multiple species in the chosen study area have shown much availability of nutrients and microbial biomass.

- L10-316: Consider delving deeper into the implications of this study.

  **Justification:** Thank you for your comment, the line 310-316 states that the present study has provided some insights into microbial diversity within different agricultural systems, further research is needed to thoroughly investigate and understand the full extent of microbial diversity across these systems

**Tables:**

- Consider including seasons in Table 2 for non-textural parameters.

  **Justification:** Thank you for your comment, the point is noted and addition will be made in the manuscript

- Add statistical analysis results (seasons, cropping systems, regions, depths) to the tables.

  **Justification:** Yes, for almost all tables, statistical analysis has been taken care of; if. In table 2 seasons, if added, then for it, statistical analysis will be carried out.